# The Unmet Promise of Synthetic Training Images: Using Retrieved Real Images Performs Better

**Scott Geng**♣    **Cheng-Yu Hsieh**♣    **Vivek Ramanujan**♣    **Matthew Wallingford**♣
**Chun-Liang Li**♣    **Pang Wei Koh**[*]♣♠    **Ranjay Krishna**[*]♣♠

♣University of Washington   ♠Allen Institute for AI
sgeng@cs.washington.edu

## Abstract

Generative text-to-image models enable us to synthesize unlimited amounts of images in a controllable manner, spurring many recent efforts to train vision models with synthetic data. However, every synthetic image ultimately originates from the upstream data used to train the generator. Does the intermediate generator provide additional information over directly training on relevant parts of the upstream data? Grounding this question in the setting of image classification, we compare fine-tuning on task-relevant, targeted synthetic data generated by Stable Diffusion—a generative model trained on the LAION-2B dataset—against finetuning on targeted real images retrieved directly from LAION-2B. We show that while synthetic data can benefit some downstream tasks, it is universally matched or outperformed by real data from the simple retrieval baseline. Our analysis suggests that this underperformance is partially due to generator artifacts and inaccurate task-relevant visual details in the synthetic images. Overall, we argue that targeted retrieval is a critical baseline to consider when training with synthetic data—a baseline that current methods do not yet surpass. We release code, data, and models at https://github.com/scottgeng00/unmet-promise.

## 1   Introduction

The success of modern machine learning systems fundamentally relies on the quantity [33, 10, 28, 57, 51, 65], quality [20, 72, 45, 44, 36], and distribution [17, 22, 62, 70, 9] of the data they are trained on. However, acquiring large amounts of quality data remains challenging, due to the sheer cost of data collection and annotation. As demand for training data continues to rise, the field is actively exploring approaches to automatically curate data at scale [20, 2, 18]. One burgeoning approach is to source *synthetic training data* from conditional generative models. Generative models enable data to be tailored to specific requirements and generated at scale, presenting a promising alternative to the challenges of real data curation. Recent work highlights this potential: for example, in natural language processing (NLP), researchers prompt strong proprietary language models to cheaply synthesize large-scale datasets for instruction tuning [68, 29, 61].

Analogously, in computer vision—the focus of our research—many recent works train models on synthetic images from modern text-to-image generators, aiming to achieve state-of-the-art visual recognition performance [56, 63, 3, 23, 24]. For example, SynCLR [62] cleverly prompts Stable Diffusion for synthetic images tailored to pre-specified downstream image recognition domains; they find that a CLIP-like model trained from scratch on the resulting *targeted* synthetic images can outperform CLIP trained on LAION-2B, a significantly larger untargeted dataset of real images. This result is quite surprising. Stable Diffusion is *also* trained on LAION-2B, so by the data processing inequality, the synthetic images it generates cannot contain any additional information about LAION-

---

[*]Equal advising.

38th Conference on Neural Information Processing Systems (NeurIPS 2024).

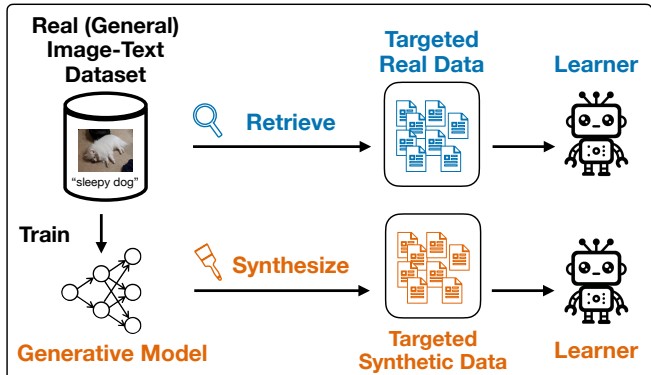

Figure 1: Given an upstream dataset of general real image-text pairs, we aim to curate a *targeted* dataset to train a learner on some target task. We can either (1) **retrieve targeted real images** directly from the upstream dataset, or we can (2) first train an intermediate generative model and then **synthesize targeted synthetic images**. By comparing these two approaches, our paper seeks to measure what value training on generated synthetic data adds.

2B over the original real LAION-2B images. Yet, training on these derivative synthetic images appears to outperform training directly on LAION-2B. How do we make sense of these additional gains? **Does the generative model truly add useful information on top of its pretraining data?**

In this paper, we argue that the performance gained from training on generated synthetic images needs to be contextualized against a critical baseline often missing in prior work: training on real images subselected from the generative model's pretraining data. In particular, prior work has often compared *task-targeted* synthetic images to *general, untargeted* real images (e.g., full LAION-2B), thereby entangling the effects of training on synthetic versus real images with the effects of targeted versus general data collection. However, these variables are not intrinsically conflated. Any generative model we use to synthesize images fundamentally derives from its upstream training data. Instead of using that upstream data to train an intermediate generative model and synthesize targeted synthetic images, we can alternatively seek to directly identify targeted real images from the upstream source through retrieval (Figure 1). By comparing synthetic training data against this retrieval baseline, we isolate the value added by the generative model.

We formalize our study under the ubiquitous problem of task adaptation, where we seek to curate task-targeted images to finetune a pretrained vision model. We empirically compare training on targeted synthetic images generated from Stable Diffusion 1.5—a text-to-image model trained on the upstream LAION-2B dataset—against training on targeted real images retrieved from LAION-2B itself. We perform hundreds of experiment runs across an order of magnitude of data scales on five visual recognition tasks (ImageNet [14], Describable Textures (DTD) [12], FGVC-Aircraft [41], Stanford-Cars [35], and Oxford Flowers102 [46]) where training on synthetic data has shown promise [62, 23].

Together, we find that **training on targeted real data retrieved from a generative model's upstream training dataset outperforms training on synthetic data from the generative model.** For example, while training on targeted synthetic images can improve downstream accuracy by up to 7.1% (absolute) on its best-case benchmark (FGVC-Aircraft), training on targeted real images helps even further, boosting accuracy by a massive 17.7%. On other benchmarks, such as ImageNet, we find that training on synthetic images can sometimes *hurt* performance even when training on real data improves it. We further show that these findings hold across several different versions of Stable Diffusion, as well as when we train on a mix of synthetic and real data. Our analysis suggests that the consistent underperformance of models trained on synthetic images is partially due to low-level generator artifacts in the synthetic images (e.g., blurs), and partially because synthetic images may distort high-level class-specific visual details that real images preserve.

Overall, we conclude that retrieval is a critical baseline to consider when evaluating the true added utility of generated synthetic training data. Our goal is not to make normative claims about whether synthetic data will ever surpass this standard, but to contribute a simple baseline to aim for, and a clean set of experiments to explicitly measure progress towards surpassing it. For instance, by conceptualizing retrieval from a generator's training data as a strong alternative to synthesizing data, a natural future direction for improving synthetic training data is to synthesize image compositions that are explicitly absent from the generator's upstream training set. Images generated in this manner may offer unique value beyond what can be retrieved from the training data. Finally, in settings where the upstream dataset of a generator is unavailable altogether (e.g., due to privacy concerns, due to proprietary data, or due to download bandwidth restrictions), the retrieval baseline is unrealizable by assumption; synthetic data therefore retains strong utility for distilling knowledge from generative

models and for privacy preservation. We release all code, models, and over 1TB of generated images to guide future work (https://github.com/scottgeng00/unmet-promise).

## 2 Related Work

**Learning from synthetic data.** Synthetic data has been widely explored in the context of many machine learning problems [68, 29, 25, 32, 61, 58, 5, 40]. For example, in NLP, synthetic data generated from strong large language models [47, 11] has been used to distill instruction-following behavior [68] and task-specific knowledge [30] into smaller models. In computer vision, prior works have sought to use synthetic data to improve the state-of-the-art across a breadth of visual tasks, such as object detection [48, 31], semantic segmentation [54, 52, 8], and optical flow [15]. Traditionally, this synthetic training data has been sourced from expert-crafted simulation and rendering pipelines [49, 15, 52, 8, 54, 31]. Recent advances in text-to-image synthesis via diffusion models [59, 27, 53] are changing this paradigm, inspiring a new wave of works that seek to train visual models on synthetic data algorithmically sampled from large-scale generative models [24, 63, 56, 23, 74, 37]. Recent works have also sought to algorithmically sample synthetic data from diffusion models trained on data-scarce domains [55]. This structural shift in the source of synthetic images from *expert-supervised programmatic simulation* to *a learned generator* that itself derives supervision from upstream data raises a critical question: does the intermediate step of training a generator and sampling synthetic data provide any added useful information over simply training on relevant parts of the upstream data directly? Our work formalizes and empirically grounds this question, contributing experiments and baselines to rigorously measure the benefits of training on modern data-derived synthetic data.

**Adapting pretrained vision models.** Large-scale pretrained image models such as CLIP [50, 10] offer transferable visual features that benefit a wide range of downstream vision tasks. It is now common practice to use pretrained models as a starting point when deploying downstream task-specific models instead of training them from scratch [69, 66]. From an algorithmic perspective, many methods have been proposed to adapt CLIP models to downstream tasks, each with varying trade-offs [71, 73, 42, 21, 4, 69]. We choose to study simple full-finetuning, centering our work on the *data* we adapt on as opposed to the algorithm. In particular, the quality and relevance of adaptation data has a crucial impact on downstream task performance; distribution shifts at inference time can significantly hurt performance [34]. Acquiring task-targeted data thus remains an active area of research [39, 67, 25]. Most related to our work is [39, 67], who also employ retrieval as a technique for collecting task targeted-data. Our work builds upon these methods to construct baselines for systematically measuring the true added utility of model-generated synthetic training images.

**Retrieval as a baseline for synthetic data.** Recent studies [74, 7] similarly use retrieval from generative model training data as a baseline for synthetic training images. For example, [74] adapts Stable Diffusion with real medical images and then generates synthetic medical images to train a ResNet from scratch; the resulting ResNet outperforms a ResNet trained on LAION-retrieved medical images. Consistent with our work, [74, 7] find that in the low-data regime of general image recognition tasks (e.g., subsampled ImageNet-1K, STL-10), augmenting real datasets with LAION-retrieved images outperforms augmenting with Stable Diffusion generated images when training ResNets from scratch. [7] further finds that this performance gap in the low-data regime persists across many synthetic image generation methods, including when they finetune Stable Diffusion with task-specific real data. In the medium-data regime (e.g. full ImageNet-1k), [7] finds augmenting with synthetic data matches but does not outperform augmenting with retrieved data. Our work generalizes the shared motivation of measuring the utility of synthetic training images to the modern data-rich regime and a wide range of visual classification tasks.

## 3 Problem Setting and Method

Given a large dataset $\mathcal{D}$ of general real image-text pairs and a downstream visual classification task specified as a set of text class names $\mathcal{C}$, we aim to algorithmically curate a **targeted adaptation dataset** $\mathcal{D}_{\mathcal{C}}$ of images $x_i$ and one-hot class labels $y_i$ to finetune and improve a pretrained vision model's performance on the downstream task. We compare two high-level approaches for sourcing this targeted data, shown in Figure 1: (1) we **retrieve targeted real images** directly from $\mathcal{D}$, forming a targeted retrieved dataset $\mathcal{D}_{\mathcal{C}}^{\text{(retrieved)}} \subset \mathcal{D}$. Alternatively, (2) we **generate targeted synthetic images**

by prompting an intermediate text-to-image generative model trained on $\mathcal{D}$, forming a targeted synthetic dataset $\mathcal{D}_{\mathcal{C}}^{(\text{synthetic})}$. We detail each approach below.

## 3.1 Sourcing data by generating synthetic images

We follow SynCLR [62], a representative method for curating synthetic training data from off-the-shelf text-to-image models. Given the set of visual class names $\mathcal{C}$, we first synthesize a large corpus of corresponding image captions by prompting a large language model (details in Appendix C.1). For example, if the class name $c \in \mathcal{C}$ is "rose," then a generated caption might be "a close-up of a pink rose in bloom." We then use these captions as input for a text-to-image generator $G$ trained on the upstream data $\mathcal{D}$, yielding a large set of synthesized images $\widetilde{x}_i$. Each image is assigned a class label $y_i$ based on the class name $c$ used to synthesize its caption. These synthetic images and labels $\{(\widetilde{x}_i, y_i)\}$ form our curated dataset $\mathcal{D}_{\mathcal{C}}^{(\text{synthetic})}$.

## 3.2 Sourcing data by retrieving real images

Rather than querying a generator trained on an upstream dataset $\mathcal{D}$, we can directly train on parts of $\mathcal{D}$ itself by retrieving relevant data. $\mathcal{D}$ consists of image-text pairs $(x_i, t_i)$. To retrieve relevant pairs, we consider two strategies. We additionally deduplicate all retrieved images with respect to our evaluation datasets following [20] to minimize test set leakage. We apply NSFW filtering [57].

**Strategy 1: hard substring matching.** Inspired by [67], we retrieve the set $\mathcal{D}_{\mathcal{C}}^{(\text{retrieved})}$ of all images $x_i$ whose corresponding caption $t_i$ contains at least one target class name $c \in \mathcal{C}$ as a substring:

$$\mathcal{D}_{\mathcal{C}}^{(\text{retrieved})} = \{(x_i, y_i) \, : \, (x_i, t_i) \in \mathcal{D} \text{ such that some class } c \in \mathcal{C} \text{ is a substring of } t_i\} \, .$$

Here, label $y_i$ is assigned based on the class $c$ contained in $t_i$. If an image-text pair $(x_i, t_i) \in \mathcal{D}$ has text $t_i$ containing multiple class names $c, c' \in \mathcal{C}$, then we simply retrieve $x_i$ multiple times and assign each instance a different label, once for each unique matched class name.

**Strategy 2: semantic $k$-NN retrieval.** Hard substring matching is simple and effective when the target visual concepts $c \in \mathcal{C}$ are concrete entities that are likely to be described in text captions (e.g., $c =$ "fire lily"), but may be less effective when the concepts are abstract (e.g., $c =$ "lined texture"). Thus, we also consider semantic (soft) retrieval via CLIP image-text embedding space similarity[2]. We convert each target class name $c \in \mathcal{C}$ into a set of natural language search queries $Q_c$ based on the templates from the original CLIP paper [50]. For each query $q_c \in Q_c$, we use approximate $k$-NN search [16] to retrieve the set $S_{q_c}$ of $k$-nearest image-text pairs $(x_i, t_i) \in \mathcal{D}$ by CLIP similarity between the query $q_c$ and *either* the image $x_i$ or the text $t_i$:

$$S_{q_c} = \left\{ \underset{(x_i,t_i) \in D}{\arg \text{top-k}} \, \text{CLIP}(x_i, q_c) \right\} \cup \left\{ \underset{(x_i,t_i) \in D}{\arg \text{top-k}} \, \text{CLIP}(t_i, q_c) \right\}.$$

We assign each image-text pair $(x_i, t_i) \in S_{q_c}$ a class label $y_i$ based on the class name in query $q_c$. We form the targeted dataset $\mathcal{D}_{\mathcal{C}}^{(\text{retrieved})}$ by unioning over all queries $q_c \in Q_c$ and all classes $c \in \mathcal{C}$:

$$\mathcal{D}_{\mathcal{C}}^{(\text{retrieved})} = \bigcup_{c \in \mathcal{C}} \bigcup_{q_c \in Q_c} \{(x_i, y_i) \colon (x_i, t_i, y_i) \in S_{q_c}\} \, .$$

## 3.3 Additional data filtering and postprocessing

Data filtering has been shown to improve training performance for both real and synthetic data and is widely used [20, 24]. We filter both our synthetic and retrieved datasets following current best filtering practices for synthetic image data [24]. Given a curated dataset $\mathcal{D}_{\mathcal{C}}$, we compute the CLIP similarity of each $x_i \in \mathcal{D}_{\mathcal{C}}$ with text corresponding to its assigned label $y_i$ (e.g., "a photo of {class name}"), constructed using the CLIP zero-shot classification templates [50]. When there are multiple templates for a given class, we aggregate by taking the maximum similarity across templates. We

---

[2]We perform semantic retrieval using precomputed LAION-2B embeddings from OpenAI CLIP ViT-L/14 [50], the same model Stable Diffusion uses to embed text prompts during generation.

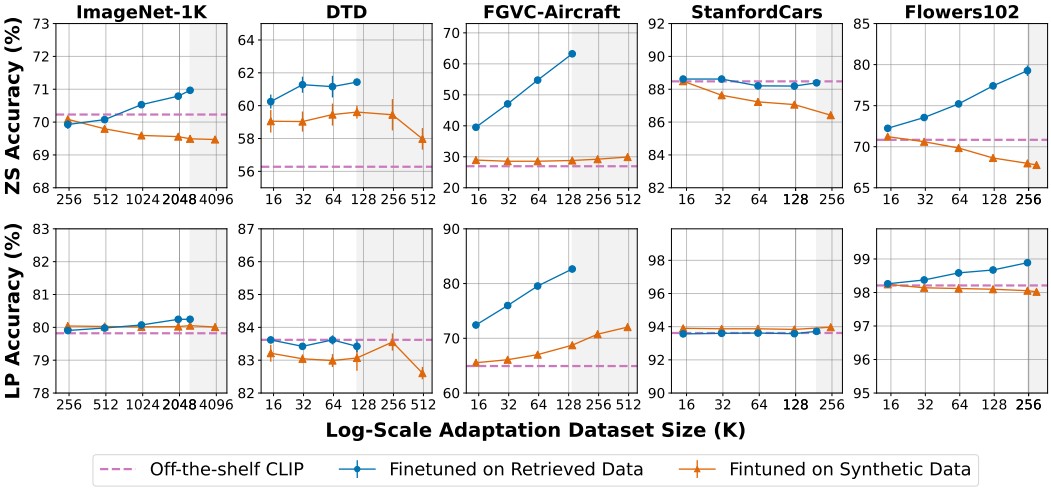

Figure 2: We adapt a pretrained CLIP image encoder (**dashed purple line**) to different downstream image classification tasks, using either (a) targeted synthetic data (**orange triangles**) generated from a Stable Diffusion model trained on LAION-2B or using (b) targeted real data (**blue circles**) directly retrieved from LAION-2B. We measure performance via downstream **zero-shot (ZS)** and **linear probing (LP)** accuracy, aggregating results over at least 3 seeds (error bars indicate $\pm 1$ standard deviation). Overall, while adapting CLIP with targeted synthetic data can sometimes improve performance over an off-the-shelf model, synthetic data is universally outperformed or matched by targeted real data. This gap persists even when we scale the sample size of the synthetic adaptation dataset beyond the maximum amount of (finite) targeted real data considered (**gray shaded regions**).

keep the top 30% of images per class by aggregate similarity. Intuitively, filtering helps remove generated and retrieved images with class-misaligned content. For example, an image labeled "dog" but without any dogs present (i.e., due to retrieval or generation errors) would receive a lower CLIP similarity score and likely be discarded. See Appendix C.2 for further discussion.

Our synthetic adaptation datasets $\mathcal{D}_{\mathcal{C}}^{(\text{synthetic})}$ are class-balanced by construction (i.e., we uniformly generate images for each class). We further postprocess the retrieved adaptation datasets $\mathcal{D}_{\mathcal{C}}^{(\text{retrieved})}$ to improve class balancing by manually fixing a global threshold $M$ and truncating the dataset such that each class label $y_i$ occurs at most $M$ times.

## 4 Main Experiments

We seek to measure the utility of learning from model-generated synthetic images. Grounding this question empirically, our experiments compare finetuning a pretrained CLIP model on (1) targeted synthetic images $\mathcal{D}_{\mathcal{C}}^{(\text{synthetic})}$ to (2) finetuning on targeted retrieved real images $\mathcal{D}_{\mathcal{C}}^{(\text{retrieved})}$.

**Benchmarks.** We focus on five downstream tasks where synthetic data has shown promise compared to similar scale *untargeted* real data [62]. We select (a) ImageNet-1K [14] and Describable Textures (DTD) [12] to evaluate recognition performance on broad categories and (b) FGVC-Aircraft [41], StanfordCars [35], and Oxford Flowers102 [46] to evaluate performance in fine-grained settings. We use standard pre-defined train, test, and validation splits when available, and otherwise randomly subset the training set to create missing train-validation splits (details in Appendix D.2).

**Finetuning data curation.** For each downstream benchmark, we first curate an adaptation dataset $\mathcal{D}_{\mathcal{C}}$ (Section 3) by either (1) generating synthetic images with Stable Diffusion 1.5 [53], trained on the LAION-2B dataset [57], or (2) retrieving real images directly from LAION-2B. We treat the choice between our substring-based and semantic retrieval strategies as a hyperparameter, using downstream validation set accuracy to determine the best choice for each benchmark. Hyperparameters for retrieval are detailed in Appendix D.1.

**Model adaptation and evaluation.** We adapt a LAION-2B pretrained CLIP ViT-B/16 [10] image encoder by finetuning on the curated adaptation dataset $\mathcal{D}_{\mathcal{C}}$ with a cross-entropy classification loss for

a pre-set number of epochs. To elucidate the scaling trends of synthetic and retrieved data, we finetune across an order of magnitude of different adaptation dataset scales, subsampled from the full targeted adaptation dataset $\mathcal{D}_\mathcal{C}$. We report zero-shot (ZS) and linear probing (LP) test set accuracy, using the benchmark train set to train the LP head. For both LP and ZS evaluation, we use the validation set to identify the best epoch and finetuning hyperparameters. For each data scale, we aggregate accuracy across the results of at least three random seeds, and report the standard deviation due to seed randomness. Please refer to Appendix D.3 for further evaluation details, and Appendix D.4 for further training and hyperparameter details.

## 4.1 Main results: synthetic training data lags behind a baseline of retrieved real images

We present our main zero-shot and linear probing scaling results in Figure 2.

**At equivalent data scales, finetuning with model-generated synthetic images can help, but is universally matched or outperformed by finetuning directly with images from the generator's training data.** Consistent with prior research [62], we find that training with targeted synthetic data can improve an unadapted model. For example, on FGVC-Aircraft—the setting where previous works have found strongest gains—finetuning with 139K Stable-Diffusion generated images improved downstream linear probing accuracy by an average of 3.8 percentage points over an off-the-shelf CLIP model ($64.9\% \rightarrow 68.7\%$); on DTD, training with 110K synthetic images improves zero-shot accuracy by 3.3 points ($56.3\% \rightarrow 59.6\%$).

However, the gains from training on synthetic data are consistently matched or surpassed by training on retrieved real data. For instance, on FGVC-Aircraft, finetuning with an equivalent 139K LAION-2B retrieved images boosts performance by a massive 17.8 points ($64.9\% \rightarrow 82.7\%$). Moreover, adapting with retrieved data can improve performance even when synthetic data does not (e.g., on ImageNet and Flowers102 zero-shot accuracy.) Finally, adapting with synthetic data can sometimes even *hurt* performance (ImageNet, StanfordCars, Flowers102 zero-shot), while targeted retrieved data improves or at least does not hurt performance across all settings considered. Given equal amounts of targeted retrieved and synthetic data, retrieved data is the clear winner.

**Synthetic data can sometimes decrease the gap with retrieved data given additional scale, but remains behind.** The amount of data we can retrieve is fundamentally limited by the finite upstream data pool. For example, even after searching all 2 billion LAION-2B samples for images containing an FGVC-Aircraft class name in the caption, substring-based retrieval returned only 139K targeted images post-filtering. In contrast, it is straightforward to create ever-larger synthetic datasets by simply generating more data.

Scaling the synthetic adaptation dataset size beyond the amount of retrieved data considered (illustrated in the gray-shaded regions of Figure 2), we find that increasing the amount of targeted synthetic data does not always improve performance. For example, on DTD, synthetic data exhibits U-shaped scaling, with performance positively scaling up to 110K synthetic training images, after which performance declines. On ImageNet, Flowers102, and StanfordCars, increasing the synthetic dataset size consistently hurts zero-shot accuracy and has minimal impact on linear probing performance.

On Aircraft, scaling helps; there is a log-linear relationship between the size of the synthetic adaptation dataset and downstream linear probing accuracy (*e.g.*, scaling from 139K $\rightarrow$ 250K synthetic images improves linear probing accuracy from $68.7\% \rightarrow 70.7\%$). However, synthetic data still lags behind retrieved data: matching the performance of a mere 15K retrieved aircraft images requires scaling the synthetic dataset to 500K images, reflecting a $\sim$33x difference in dataset size and required finetuning compute. Naively extrapolating this ratio, matching the performance of the full 139K retrieved adaptation dataset would require nearly 5M synthetic images after top 30% filtering. We note, however, that synthetic data is unlikely to truly scale infinitely, as synthetic data fundamentally derives from the (finite) training set of our generative model. Still, the performance of synthetic data is likely unsaturated at the 500K scale (*i.e.*, accuracy is still trending up); due to compute limitations, studying whether further scaling can outperform retrieved data is left for future work.

**Synthetic data can improve a model's task representation without significantly improving the model's task performance.** Broadly speaking, zero-shot task accuracy measures a model's ability to directly solve the downstream task, whereas linear probing accuracy measures the quality of the model's learned *task-relevant representation*. We find that even when training on synthetic data improves the model's representation (i.e., downstream linear probing accuracy), it may not

significantly improve the model's zero-shot accuracy. In contrast, when training on retrieved data improves the model's representation, zero-shot accuracy also exhibits positive scaling. For example, CLIP adapted with either 15K retrieved images or 500K synthetic images both achieve a similar linear probing accuracy ($\sim 72\%$), yet the model adapted with synthetic data achieves a much worse zero-shot accuracy ($28.9\%$ versus $39.5\%$). We discuss possible reasons for this qualitative discrepancy in model behavior in our analyses below (Section 5.1).

## 5 Analysis

In this section, we explore two questions to better understand our main results. First, what factors drive the underperformance of synthetic data? Second, do our findings hold under variations of our experimental setup? We focus our analysis experiments on ImageNet, to understand general image recognition performance, and FGVC-Aircraft, the sole benchmark where synthetic data exhibited strong positive log-linear scaling. Additional analysis experiments are presented in Appendix B.

### 5.1 Why does synthetic data lag retrieved real data?

**Qualitative visualizations.** We visualize a random selection of images from our curated synthetic and retrieved adaptation datasets in Figure 3. Compared to retrieved real images, we observe that the synthetic images (1) contain low-level generator artifacts, and (2) differ in visual content distribution, both in terms of semantic details and overall image composition. For example, although the synthetic FGVC-Aircraft adaptation images (top two rows of Figure 3) are recognizable as airplanes, the visual content often contains incorrect class-relevant semantic details: a correctly-depicted "Airbus A320" should have one engine per wing and two sets of wheels at its rear, yet our synthetic images often exhibit incorrect engine or wheel configurations. This qualitative discrepancy in visual detail precision may partially explain why training on synthetic data does not improve task zero-shot accuracy; synthetic images do not retain enough class-accurate details to directly teach the model the downstream task. In contrast, training on synthetic images can improve linear probing accuracy, because the synthetic images still broadly look like aircraft and thus may help align the model's representation to the downstream domain.

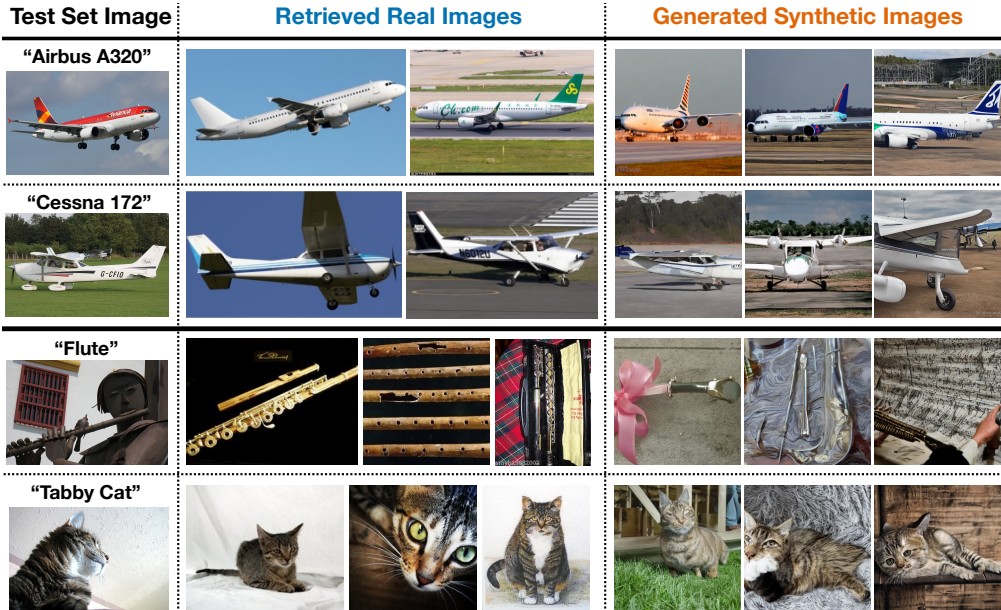

Figure 3: We visualize **retrieved real images** and **synthetic images** from our targeted adaptation datasets for FGVC-Aircraft (top two rows) and ImageNet-1K (bottom two rows), alongside ground truth images (left column) for reference. Compared to retrieved images, synthetic images often (1) contain generator artifacts (e.g., the blur on the edges of the "Cessna 172", the eyes and mouth of the "Tabby Cat") and also (2) distort class-relevant visual content, such as the engine configuration of a true "Airbus A320" (i.e., exactly one engine per wing) and the entire visual appearance of a "Flute". We hypothesize that both factors contribute to synthetic training data's underperformance versus real training data.

**Synthetically perturbing retrieved real images.** To disentangle the effect of low-level generator artifacts and visual content differences between synthetic and retrieved real images on downstream model performance, we trained on "hybrid" images that have similar semantic visual content as our retrieved real images but contain generator artifacts like our synthetic images. Following SDEdit [43], we use Stable Diffusion to synthetically perturb our retrieved images to introduce model-specific artifacts present in the synthetic images Stable Diffusion generates. Given a noise strength parameter $\gamma \in [0, 1]$ and a retrieved image $x_0$, SDEdit adds Gaussian noise to $x_0$ according to timestep $t = \gamma$ of Stable Diffusion's time-dependent forward process. We then denoise the noisy image using the same reverse diffusion process as in text-to-image generation, yielding a perturbed image $x^{(\gamma)}$ that looks semantically like $x_0$ while also containing Stable Diffusion-specific artifacts. Increasing $\gamma$ increases the amount of Gaussian noise added to $x_0$, thereby increasing the severity of visual artifacts introduced in the resulting $x^{(\gamma)}$ (see Appendix E.1 for further details). In pseudocode,

$$x^{(\gamma)} = \text{StableDiffusion.denoise}(\text{StableDiffusion.add\_noise}(x_0, \gamma), \gamma).$$

Starting from the full targeted retrieved adaptation datasets $\mathcal{D}_{\mathcal{C}}^{(\text{retrieved})}$ for FGVC-Aircraft and ImageNet, we use SDEdit to introduce generator artifacts into the retrieved real images over a range of $\gamma$ values. We visualize the resulting perturbed images in Figure 4, and plot the results of training on these perturbed images across $\gamma$ in Figure 5.

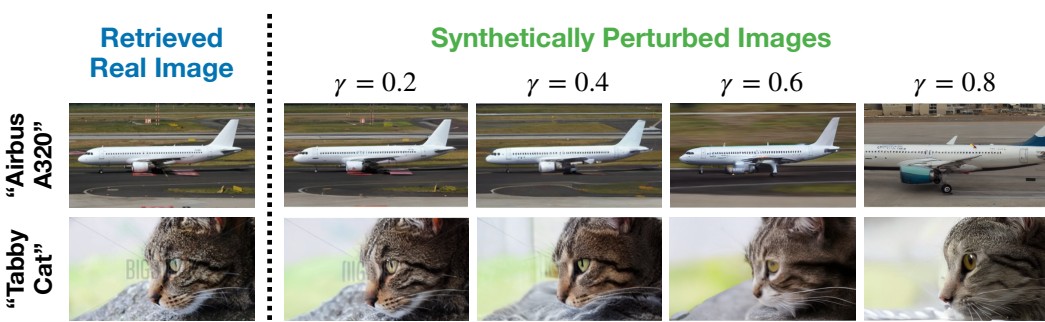

Figure 4: We use Stable Diffusion to synthetically perturb real images according to a noise strength parameter $\gamma \in [0, 1]$, where larger $\gamma$ increases the severity of generator-specific artifacts added by the perturbation. When $\gamma \geq 0.6$, the introduced artifacts can be strong enough to damage task-relevant visual details for finegrained tasks like FGVC-Aircraft (e.g., the airplane's engine and rear wheels). For broad tasks like ImageNet, artifacts have a lesser impact on class-relevant details; the "Tabby Cat" is recognizable as a cat even after perturbing with high $\gamma$.

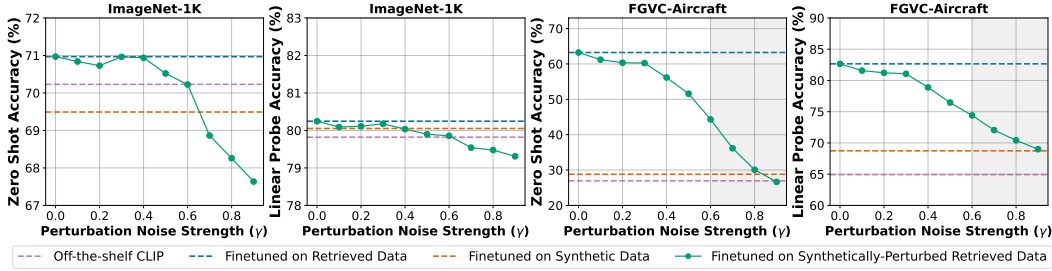

Figure 5: We finetune a pretrained CLIP model (**dashed purple line**) on retrieved real images that are synthetically perturbed (**green circles**) with Stable Diffusion to introduce generator artifacts. The perturbation strength is controlled by a parameter $\gamma \in [0, 1]$ where larger $\gamma$ introduces stronger artifacts; within the **gray-shaded region**, the artifacts are strong enough to damage class-relevant details. Our results suggest that generator artifacts do contribute to synthetic data's underperformance—any artifact level causes performance to drop below training on retrieved images (**dashed blue line**). Moreover, differences in visual content between synthetic and retrieved images also matter; even with relatively strong perturbations ($\gamma = 0.5$), training on artifact-afflicted perturbed images that retain the semantic content of retrieved images outperforms training on synthetic images (**dashed orange line**).

Our results suggest three takeaways. First, generator artifacts indeed contribute to the underperformance of synthetic training images, especially for fine-grained classification tasks. On FGVC-Aircraft, any amount of added generator artifacts drops downstream accuracy. Second, the impact of artifacts is relatively lower for broad classification domains such as ImageNet, where downstream performance is not significantly impacted until we perturb with a relatively strong noise strength of $\gamma = 0.5$. Finally, visual content differences between synthetic and retrieved images also play a key role in the performance gap between synthetic and retrieved training data. When we perturb images with strength $\gamma = 0.5$, the resulting images are heavily afflicted with artifacts, but still retain the important class-relevant details of retrieved real images, such as correct airplane engine configurations. Training on $\gamma = 0.5$ images significantly outperforms training on synthetic images. Intriguingly, training on aircraft images perturbed beyond the point where class-relevant visual details are damaged ($\gamma \geq 0.6$) still outperforms synthetic data; we speculate that this is because these heavily perturbed images still retain the overall image composition of retrieved images.

## 5.2 Synthesizing data via another generative model

For our main scaling experiments, we generate synthetic image datasets using Stable Diffusion 1.5 to maintain consistency with prior work [62, 23]. To what degree does our choice of generative model impact our findings? At the time of our study, Stable Diffusion 1.x models are the only modern text-to-image models with open source training data available to retrieve from. Therefore, we focus our study here on Stable Diffusion (SD) versions 1.1, 1.3, and 1.5. Starting from SD v1.1, which is trained on the full LAION-2B dataset, SD v1.3 and v1.5 are derived by further finetuning on high-quality subsets of LAION-2B. This additional finetuning improves image generation fidelity [53], but may lead to the model forgetting parts of the LAION-2B distribution [19]. Following our main experiment setup (Section 4), we use SD v1.1 and SD v1.3 to generate various-sized targeted synthetic adaptation datasets for ImageNet and FGVC-Aircraft. Results are plotted in Figure 6. Overall, while training with synthetic data from different generative models yields varying performance, synthetic data from all generative models considered consistently fall short of retrieval.

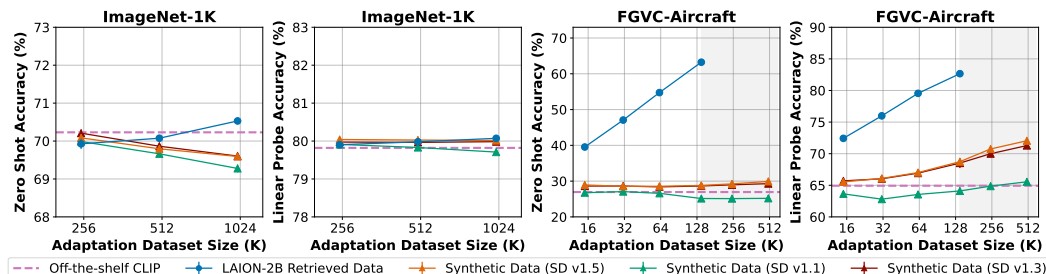

Figure 6: We ablate our choice of generative model used to synthesize images in our main experiments, comparing training on synthetic images from **Stable Diffusion 1.5 (SD v1.5)** to synthetic images from **SD v1.1** and **SD v1.3**. Across multiple Stable Diffusion models—the only modern text-to-image generators with open-source training data available—training with images directly retrieved from the generative models' training dataset (**LAION-2B**) outperforms training with generated images.

## 5.3 Mixing synthetic and retrieved data

On FGVC-Aircraft, finetuning CLIP with either synthetic or retrieved data alone consistently improves downstream task accuracy. The gains from retrieved data are stronger than the gains from synthetic data across all data scales; however, synthetic data may improve CLIP in ways that is complementary to retrieved data, and thus present orthogonal value. To test this possibility, we measure whether training on a mix of synthetic and retrieved Aircraft adaptation data significantly outperforms training with either alone. Starting from our largest retrieved adaptation dataset $\mathcal{D}_{\mathcal{C}}^{(\text{retrieved})}$ (139K images), we progressively add in increasing amounts of synthetic images from our synthetic adaptation dataset $\mathcal{D}_{\mathcal{C}}^{(\text{synthetic})}$ and finetune a pretrained CLIP model with the resulting mix. We plot results in Figure 7. We find that training on the mixed images outperforms training on synthetic images alone; however, training on a mix significantly drops performance compared to using retrieved data alone.

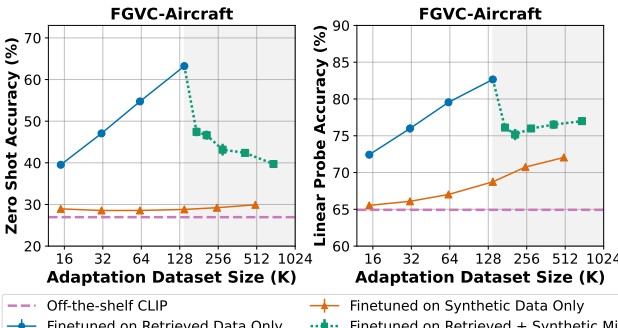

Figure 7: We finetune a pretrained CLIP model (**dashed purple line**) on a dataset comprising a mix of targeted synthetic and retrieved real images (**green squares**). The mix is constructed by adding varying amounts of synthetic images to our full retrieved adaptation dataset. Training on the mix outperforms training on synthetic images alone (**orange triangles**), but is strictly worse than training on retrieved images alone (**blue circles**).

## 6 Discussion

Our work sought to answer a key question: given that all model-generated synthetic images derive from the generator's upstream training data, does training on synthetic images provide value over training on the parts of the upstream real data directly? We contribute a set of rigorous experiments to ground this question empirically, and discover that training on upstream real images collected via our simple retrieval baseline significantly outperforms training on synthetic images. Our initial question is answered negatively. We therefore argue that retrieval is a critical baseline to surpass in order to show value from synthetic training data, and encourage comparison against it in future research.

Importantly, we do not seek to make normative claims about whether training with synthetic images will ever surpass this baseline—future work may unlock gains that we have not yet found. As a first step, we contribute analyses of why synthetic training images underperform upstream real images, finding that both generator artifacts and semantic errors within synthetic images are key areas for future improvement. Furthermore, given that image retrieval is a strong alternative to image synthesis, a natural next step is to generate image compositions that are explicitly rare or absent from the generator's upstream training dataset; we are optimistic that synthesizing these "missing" images may offer unique value beyond what is present in the existing upstream real images. Such an approach leverages the compositional generalization abilities of the generator, which recent research promisingly suggests may be stronger than the compositionality of a discriminative model trained on the same upstream data [38, 13].

Finally, our findings assume access to the generative model's upstream training data, an assumption that may not always hold. The upstream pool may be proprietary or strictly regulated due to privacy concerns. In such settings, training directly on the upstream data is impossible; synthetic data from a generative model trained on this unavailable upstream data remains an exciting alternative to acquire otherwise inaccessible information.

**Limitations.** As an empirical study, our compute budget limits the number of experimental variations we consider. Our results are derived from adapting CLIP models with standard full finetuning; we conjecture that our findings generalize to other large-scale pretrained backbones and adaptation methods as well, but we were not able to test this empirically. Moreover, at the time of our work, Stable Diffusion is the only text-to-image model with publicly available training data to retrieve from (i.e. LAION-2B); we do not study other generators trained on other data pools. Finally, we focus on model accuracy, leaving a comparison of model robustness and fairness from training on synthetic versus real data to future work.

## Acknowledgments and Disclosure of Funding

We graciously thank (in alphabetical order) Eric Frankel, Jacqueline He, Athena Tsu, and Rui Xin for their helpful comments and feedback. SG is supported by the NSF GRFP. PWK is supported by the Singapore National Research Foundation and the National AI Group in the Singapore Ministry of Digital Development and Innovation under the AI Visiting Professorship Programme (award number AIVP-2024-001).

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

## A Broader Impacts

We release all models and synthetic data from our work to benefit future research. All assets released are meant to be scientific research artifacts. Moreover, all models released are task-specific classification models, limiting potential for misuse. Nonetheless, we do not encourage the use or deployment of our models in practice.

## B Additional Findings

We further validate the main findings of our paper by (1) providing further analysis of the synthetic training data and (2) considering two additional variations upon our experimental setup: applying more aggressive decontamination, and generating synthetic images with alternative prompting strategies.

### B.1 Additional data analysis: CLIP score distributions

Do the distributions of CLIP similarity scores in the final synthetic and retrieved training datasets significantly differ? Since we filter the synthetic and retrieved data independently of each other to maintain a clean experimetnal setup (i.e., we do not use any information from the retrieved images to inform the selection of synthetic data, and vice versa), there may be differences in CLIP similarity that potentially explain gaps in downstream training performance.

We histogram the CLIP similarity score distributions of the resulting filtered synthetic and retrieved training data in Figure 8. Overall, despite setting the filter score threshold for synthetic and retrieved data independently, we find that the distribution of post-filtering synthetic image CLIP scores is right-shifted compared to the distribution of post-filtering retrieved image CLIP scores. In other words, synthetic images have comparable or even higher CLIP scores than retrieved images on average; CLIP judges synthetic data to be higher quality on average. CLIP score differences alone do not explain the lagging training performance of synthetic data.

Further corroborating this finding with a case study, we observed that CLIP assigns the synthetic "flute" images in Figure 3 relatively high similarity scores of $0.285$, $0.265$ and $0.263$, despite the fact they they are obviously wrong to the human eye. For reference, a CLIP score of $0.249$ reflects the top $30\%$ of all retrieved 'flute' images. Overall, while the current best practice of using CLIP score to filter synthetic data does improve performance, CLIP score remains limited. Future work may explore synthetic data filtering methods that do not depend on CLIP.

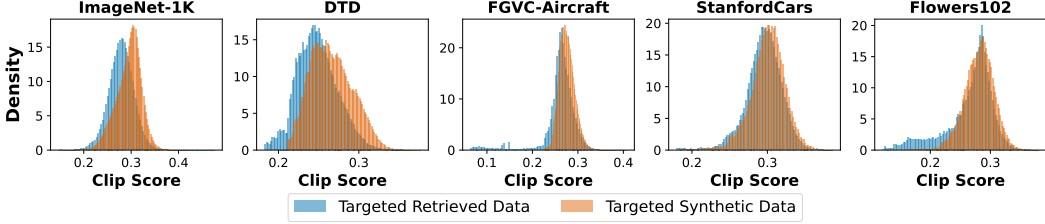

Figure 8: We visualize the distribution of CLIP scores in the **filtered retrieved** and **filtered synthetic** targeted adaptation datasets. Overall, CLIP assigns synthetic data higher scores on average, despite its lower training performance.

### B.2 Additional experiment: further decontaminating retrieved data for benchmark train sets

For our main experiments, we decontaminated all retrieved data for the downstream benchmark test sets to avoid test set leakage. However, retrieving from LAION-2B may also result in images from downstream benchmark training sets being included in the retrieved data. While this contamination will also affect synthetic data, as the generator is trained on the same data we retrieve from, the potential contamination is arguably more explicit when we retrieve directly. We thus further decontaminated all retrieved data for the benchmark train sets following [20]. We report the amount data removed by this additional decontamination step in Table 1 and plot the results of training on

twice-decontaminated retrieved data in Figure 9. Overall, while some retrieved images were indeed similar to train set data (an average 1.9% of retrieved data was removed), discarding them minimally impacted performance. Our main findings remain unchanged.

| ImageNet-1K | DTD | FGVC-Aircraft | StanfordCars | Flowers102 |
|---|---|---|---|---|
| 145370 (5.6%) | 0 (0.0%) | 328 (0.2%) | 6647 (3.5%) | 342 (0.1%) |

Table 1: We report the number of images removed from our original retrieved datasets (which are test set decontaminated) after further decontaminating with respect to the downstream benchmark train set. We report the percentage of the retrieved dataset removed in parentheses.

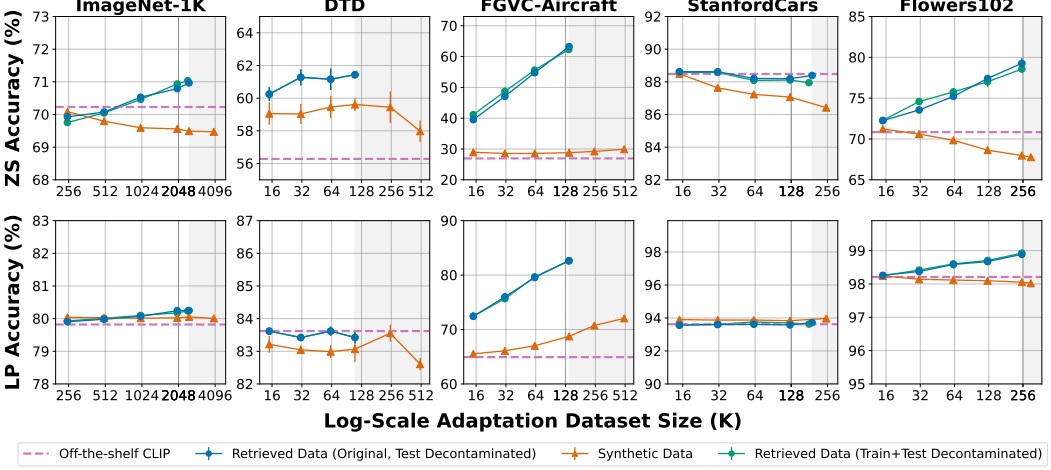

Figure 9: We further decontaminate our **original retrieved data** (already test set decontaminated) for the benchmark train set and finetune **off-the-shelf CLIP** on the **train+test decontaminated retrieved data**. Overall, train set decontamination has little performance impact; **synthetic data** remains behind. Note, no train set contamination was found for DTD.

## B.3    Additional experiment: generating synthetic data with alternative prompts

In our main experiments, we generate synthetic data via LLM-generated image captions following the SOTA method of SynCLR [62]. SynCLR shows that prompting for synthetic images with LLM captions outperforms many alternative prompting strategies at the hundred-million scale, such as prompting with alt-text from LAION images. To ensure that we utilize the most performant synthetic data generation method available, we sought to validate this finding in our experimental setup. We further compare against the strategy suggested in [37], which involves prompting with BLIP-2 generated captions of real images.

Starting from the unfiltered retrieved training data, we used BLIP-2 to caption each image and construct prompts of the form "a photo of classname, BLIP-2 image caption" following [37]. We then performed top-30% CLIP filtering on the resulting synthetic images. We compare the performance of training with filtered synthetic images generated with three prompting distinct strategies to our filtered retrieved data in Figure 10. Specifically, we compare training with filtered synthetic images generated from (1) our original LLM prompts, (2) BLIP-2 captions, and (3) LAION alt-text from retrieved data. Overall, among the three generation strategies, our original LLM prompts perform best on ImageNet, and perform comparably to BLIP-2 captions and LAION alt-text on Aircraft.

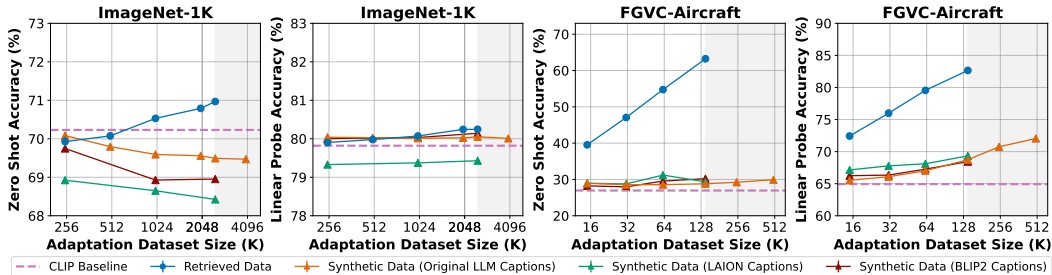

Figure 10: We compare the training performance of targeted synthetic images generated via (1) our **original LLM-guided prompts**, (2) **LAION alt-text** from **retrieved data**, or (3) **BLIP-2 captions** of retrieved data. Our original LLM prompts yield the best synthetic data for ImageNet; all 3 generation stratgies perform comparably on Aircraft. Regardless of the strategy chosen, synthetic data lags retrieved data.

# C   Details in Methodology

## C.1   Sourcing data by generating synthetic images

Given a set of visual class names $\mathcal{C}$ from our target task, we first synthesize a large corpus of image captions for each class name by prompting a large language model (we use Llama-2 7B [64]). For each concept name $c \in \mathcal{C}$, we use three type of prompts to convert $c$ into an image caption following [62]. For the sake of completeness, we detail the prompts here:

1. $c \mapsto$ caption. We prompt the language model (LM) to directly translate the class name into a caption using a prompt with 3 few-shot in-context examples.

2. $c, background \mapsto$ caption. We prompt the LM with an additional background attribute that is randomly sampled from a set that is predetermined based on the domain of $\mathcal{C}$. For example, if $\mathcal{C}$ contains a list of flower names, then possible background attributes might include "garden," "meadow," or "forest." These background attributes are automatically generated by prompting a strong instruction-tuned language model such as GPT-4 [1] with the class names $\mathcal{C}$. We provide the LM with 3 in-context examples of $c, background \mapsto$ caption mappings.

3. $c, relation \mapsto$ caption. We prompt with an additional spatial relationship attribute that is sampled from a domain-invariant set of relationships, such as "next to," "below," "besides," etc. We provide 3 in-context examples of $c, relation \mapsto$ caption mappings.

Each of these captions are directly used as text input to Stable Diffusion 1.5 to produce our targeted synthetic dataset $\mathcal{D}_{\mathcal{C}}^{(\text{synthetic})}$. When sampling from Stable Diffusion, we denoise for 50 DDIM [60] steps starting from Gaussian noise, using a classifier-free guidance [26] scale of 2.0. We choose Stable Diffusion as our generator because (1) it is pretrained on an open-source dataset, LAION-2B, and (2) to better maintain consistency with recent work [62, 63, 56, 23].

Generating the images can be computationally expensive; every 1M synthetic images generated (pre-filtering) requires around 12 hours of generation on 64 NVIDIA A100 GPUs. To lower the barrier for future research, we will release our generated synthetic images at https://github.com/scottgeng00/unmet-promise.

## C.2   Filtering synthetic and retrieved data

We rank and filter synthetic data independently, using a per-class threshold to identify and keep the top 30% of images for each class and for each dataset. We adopt CLIP as our synthetic image filter as it is the current best filtering method we are aware of [24].

# D   Details in Experimental Setup

## D.1   Retrieval hyperparameters

We perform $k$-NN retrieval with $k = 2000$ for every downstream benchmark except ImageNet-1K, where we use $k = 500$. We picked these values of $k$ with a rough target of retrieving 10K images per class. In particular, the original CLIP paper [50] has a different set of class template strings for each benchmark; our query sets $Q_c$ for each benchmark are differently sized, and our values of $k$ vary to reflect that.

For class balancing, we set $M = 10000$. We do not tune $k$ or $M$ in our experiments.

Choosing based on downstream validation set accuracy, we use our substring retrieval strategy for FGVC-Aircraft and Flowers102; we use our semantic retrieval strategy for ImageNet-1K, DTD, and StanfordCars.

We use precomputed $k$-NN search indicies from LAION-2B [57] to query against OpenAI CLIP ViT-L/14 image embeddings. No search indicies are available for querying against text embeddings; we construct our own using FAISS [16], using the configuration `OPQ256_768,IVF131072 HNSW32,PQ256x8`. Computing 2 billion OpenAI CLIP ViT-L/14 text embeddings for the captions in LAION-2B took approximately 2 hours on 100 GPUs of varying capacity. Computing the search index from the embeddings took approximately 12 hours on 128 CPU cores.

## D.2   Details for downstream benchmarks

We use the standard pre-defined train-test-validation splits for FGVC-Aircraft, DTD, and Flowers-102. Standard validation splits are not available for StanfordCars and ImageNet-1K. We construct a train-validation split for StanfordCars by randomly splitting 80% and 20% of the pre-defined training set (respectively) using `torch.utils.data.random_split` with random seed 42. We construct a validation set for ImageNet-1K by randomly subsampling 50K images from the pre-defined training set using `torch.utils.data.random_split` with random seed 42.

## D.3   Details for model evaluation

When performing zero-shot evaluation, the finetuned model $W \circ \text{CLIP}$ is directly used to inference the downstream test set without any additional tuning. When performing linear probing evaluation, we replace $W$ with a randomly initialized linear head $W'$, freeze the CLIP encoder, and train $W'$ with a standard cross entropy classification loss on the downstream benchmark train set; we perform inference with $W' \circ \text{CLIP}$.

## D.4   Details for model adaptation

**Finetuning details.**   To finetune CLIP for a specific downstream image classification task, we first initialize a linear readout head $W$ using the weights from the text-based zero-shot CLIP model [10]. Concretely, we initialize $W$ using the CLIP text embeddings of the class names for the desired downstream task. We then append the classification head $W$ on top of CLIP's vision encoder, and train end-to-end using a standard cross entropy classification loss against one-hot labels.

We could alternatively choose to finetune CLIP with a contrastive objective, where each positive pair is a synthetic or retrieved image alongside its corresponding caption. However, we find that cross entropy finetuning performs better across the board, so we use cross entropy finetuning for all experiments in our paper.

A full adaptation dataset scale sweep for a single benchmark and a fixed set of hyperparameters takes approximately 24-36 hours on 2 NVIDIA A40 GPUs.

**Random seed.**   For our main experiments, generator ablations, and data mixing experiments we report results aggregated across at least three random seeds. The random seed is used to (1) seed the training algorithm, and (2) controls adaptation dataset subsampling.

**Hyperparameter details.**   We start with relatively standard hyperparameters from prior work [69], and initially tune them in our setting by finetuning CLIP on a small-scale dataset of retrieved or

synthetic images from each downstream benchmark and grid-sweeping by learning rate and batch size. From the hyperparameters we tried at this scale, we find the following work best for both synthetic and retrieved images across all downstream benchmarks:

- Batch size: 512
- Warmup steps: 500
- LR schedule: Cosine decay
- L2 weight decay: 0.1

We find that models are sensitive to learning rate; there is no one optimal learning rate across all settings. Thus, for our full-scale experiments, we sweep learning rate across {5e-4, 1e-5, 1e-6}, and select the best learning rate for each downstream benchmark based on validation set accuracy.

We train with an AdamW optimizer, using $\beta_1 = 0.9, \beta_2 = 0.95$.

On all benchmarks except ImageNet, we finetune for a fixed 30 epochs. On ImageNet, we train for a fixed 10 epochs to save compute, as we found that validation set accuracy plateaued early on.

# E  Details in Analysis Experiments

## E.1  Synthetically perturbing retrieved images

We use SDEdit [43] to synthetically perturb the retrieved real images by adding Gaussian noise and then denoising with Stable Diffusion 1.5. We study the perturbation across 10 different choices of the noise scale (i.e. perturbation strength) parameter $\gamma \in \{0.1, 0.2, 0.3, 0.4, 0.5, 0.6, 0.7, 0.8, 0.9, 1.0\}$. All other hyperparameters for the perturbation (e.g. guidance scale, noise scheduler) are fixed to the same hyperparameters used to generate synthetic images from scratch (detailed in Appendix C.1) When using SDEdit to perturb a retrieved real image, we additionaly require a text prompt to guide the denoising. We find that using an empty string for the prompt results in incoherent images. We thus use the string "a photo of {classname}", where {classname} is the label name assigned to the retrieved image being perturbed. Thus, even at noise scale $\gamma = 1.0$ (i.e., beginning the denoising at the endpoint of Stable Diffusion's forward diffusion process[3]), we would not expect model performance from training on perturbed retrieved images to be identical to model performance from training on our synthetically generated images; our synthetically generated images are synthesized using LLM-written prompts that contain richer information.

# F  Details in Licensing Information

## F.1  Benchmarks

ImageNet-1K is released under a custom license that specifies non-commercial research use only. Details can be found at https://www.image-net.org/download.php. Licensing information is unavailable for DTD, Flowers102, FGVC-Aircraft, and StanfordCars; all four datasets are products of academic research and are publicly available online for download.

## F.2  Models

Stable Diffusion 1.1, 1.3, and 1.5 are all released under a CreativeML OpenRAIL M license. Open-CLIP models and OpenAI CLIP are released under MIT License.

## F.3  Data

LAION-2B metadata, precomputed embeddings, and $k$-NN search indices are released under CC-BY-4.0 licensing. As a web-scraped dataset, all images pointed to by the URLs retain their original licenses.

---

[3]Even at $\gamma = 1.0$, which maps input images to the endpoint of the forward diffusion process, Stable Diffusion's noise schedule does not convert the input image into pure isotropic Gaussian noise. This is due to the noise schedule parameters.

### F.4 Software

We build off the open source code of [53, 39, 6, 16, 69]. FAISS [16] and clip-retrieval [6] are released under MIT license. SynCLR code [62] is released under Apache 2.0. Stable Diffusion code [53] is released under CreativeML Open RAIL-M. WiSE-FT (the codebase we build off of for CLIP finetuning) is released under an MIT license.

