# OpenReview forum: "The Unmet Promise of Synthetic Training Images: Using Retrieved Real Images Performs Better"
_NeurIPS.cc/2024/Conference — NeurIPS 2024 poster_

### Official Review · Reviewer_6uYi · 2024-06-26

**Soundness:** 4
**Presentation:** 4
**Contribution:** 3
**Rating:** 8
**Confidence:** 4

**Summary:**

This paper studies whether training on synthetic images from generative model can **truly** surpass the baseline of training on the retrieved real images that are used to train the generative model. It provides several key insights: 1) retrieved real images are significantly superior to synthetic images across a wide range of recognition tasks, 2) both data sources (retrieved and synthetic) are beneficial to original training images, and 3) adding synthetic images to retrieved images will ruin the gain achieved by the latter. It also analyzes two factors that may cause the inferiority of synthetic images.

**Strengths:**

1. The paper is very clearly written.
2. The key insight (synthetic images are indeed inferior to the naive baseline of retrieved real images) is very interesting and timely. I believe it was previously totally overlooked by our community. There have been many works trying to utilize synthetic images for representation learning recently, but they mostly utilize synthetic images *blindly* and fail to dig into the critical question asked by this paper.
3. The analysis for the poorer performance of synthetic images is convincing, especially the ablation study on the synthetically perturbed real images, which is quite inspiring.
4. The scope and position of this paper are properly defined. It does not aim to uncover the uselessness of synthetic images, but to present a necessary baseline for future works to compare with. I believe this simple yet strong baseline will motivate future works to construct and leverage synthetic images more effectively. Besides, the authors also consider and discuss the scenarios where synthetic images are indispensable, e.g., scenarios with privacy concerns.

**Weaknesses:**

I do not think there are critical weaknesses in this paper. I only have one minor concern. See below.

**Questions:**

I love this paper, only with one minor question or suggestion. Indeed, it is somewhat expected that synthetic images may not be as good as retrieved synthetic images in standard benchmarks with widely existing class concepts. Could the authors provide more insights about the potential advantages of synthetic images in scenarios with rare or very complex concepts?

**Limitations:**

The authors have clearly discussed the limitations.

---

> ### Author Rebuttal · Authors · 2024-08-07
>
> Thank you for your valuable feedback! We are honored to hear that you enjoyed our paper, and are grateful you found our research question “timely” and “totally overlooked” by our community.
>
> Your question—whether synthetic images can offer unique advantages for tasks with rare or complex concepts—is very exciting. Such concepts are precisely the concepts for which collecting labeled real data is difficult, and thus exactly where we have the greatest need for synthetic data. These directions are exciting topics for future work that we will discuss further in our final paper. For example, synthesizing very complex concepts might be made feasible by leveraging the compositional generalization abilities of generative models, which recent research promisingly suggests may be a unique boon of generative models compared to other models trained on the upstream data [1, 2] (references below).
>
> Just for curiosity’s sake, we also performed a preliminary study to try and relate the concept-wise performance of models trained on synthetic or retrieved data with the rarity of each concept. Specifically, given a concept denoted via a text string $c$ (e.g., “Airbus A320”), we approximate that concept’s frequency in LAION-2B by counting the number of image-text pairs with text containing $c$ as a (case-insensitive) substring. We scattered the concepts with their frequency on the $x$-axis and model accuracy on the $y$-axis. Unfortunately, we were not able to find any clean trends — with existing methods, the performance gap between synthetic and retrieved data does not appear to systematically decrease on rarer concepts. To speculate, this may be because generative models are trained on the same data pool that we retrieve from, and thus may also have difficulty learning rare concepts during general pretraining as they are seen less frequently. Recent work [3] corroborates this conjecture; however, we note there are asterisks to our preliminary study, as we did not extensively validate the robustness of our concept frequency metric. Nonetheless, if off-the-shelf pretrained generators are indeed less effective at generating rare concepts, a potential future direction would be to resolve this limitation by taking advantage of a small set of examples to adapt the generative model in a sample efficient manner (e.g., perhaps through textual inversion). Afterwards, we may be able to compose the concept with the generator’s existing knowledge to sample new data variations.
>
> Thank you again for your time, and thank you for drawing our attention to these exciting future directions that may lead to gains beyond our retrieval baseline. We will discuss them thoroughly in the next version of the paper!
>
>
> * [1] Your diffusion model is secretly a zero-shot classifier. Alexander C Li, Mihir Prabhudesai, Shivam Duggal, Ellis Brown, and Deepak Pathak.
>
> * [2] Text-to-image diffusion models are zero shot classifiers. Kevin Clark and Priyank Jaini.
>
> * [3] No "Zero-Shot" Without Exponential Data: Pretraining Concept Frequency Determines Multimodal Model Performance. Vishaal Udandarao, Ameya Prabhu, Adhiraj Ghosh, Yash Sharma, Philip H.S. Torr, Adel Bibi, Samuel Albanie, Matthias Bethge.

---

> > ### Comment · Reviewer_6uYi · 2024-08-10
> > **Thank you for response**
> >
> > I go through other reviews and the authors' responses. I will keep my original score of 8. I believe this paper is worth presenting to our community since it highlights a competitive baseline in learning from synthetic images. This baseline will motivate subsequent works to more effectively explore the role of synthetic images in this new era. I think this big advantage has outweighed other minor disadvantages. The authors are also recommended to follow other reviews' advice to further polish the details of this paper.

---

> ### Author Response · Authors · 2024-08-12
> **Thank you for your feedback!**
>
> Thank you very much for your time and effort in this process! We appreciate your insightful comments, and will be sure to incorporate all other reviewer's valuable feedback in the next version of our paper. Thank you for helping make our work stronger!

---

### Official Review · Reviewer_Etje · 2024-07-11

**Soundness:** 3
**Presentation:** 3
**Contribution:** 3
**Rating:** 6
**Confidence:** 4

**Summary:**

There is a growing interest in training vision models using synthetic data. This paper explores the effectiveness of synthetic data compared to real images directly retrieved from image generator's training sets like LAION-2B. The experimental results indicate that, while synthetic data can be beneficial for some tasks, real data often matches or surpasses its performance. The paper suggests that these results are due to high-frequency generator artifacts and inaccuracies in task-relevant visual details present in synthetic data.

**Strengths:**

* This paper proposes a new baseline for training with synthetic data, which is novel and interesting.
* This paper presents extensive experiments, such as different datasets and data scales.
* The experiment of synthetically perturbed images is novel.

**Weaknesses:**

The proposed baseline using retrieved images is novel and inspiring. However, some experimental settings may limit the results of training on synthetic images, leading to unfair comparisons and potentially misleading conclusions.

- Settings regarding image quality:
  - Data filtering: The paper keeps images with top CLIP similarity as the training set (Section 3.3). Are synthetic and retrieved training data ranked and selected separately? What is the distribution of CLIP similarity in the final synthetic and retrieved training datasets?
  - Synthetic images with generation artifacts: As shown in Figure 3, synthetic 'flute' images have obvious generation artifacts. These images should be filtered for a fair comparison. What is the CLIP similarity of these images? What are the model results when training on a 'clean' dataset without these images?

- Settings regarding data distribution and variation:
  - Prompts: The current prompts for generating images may provide less information and variation than retrieved real data, influencing accuracy. How about sourcing the generation model using captions from retrieved data or other real images, as suggested in [1]?
  - Data scaling: Although synthetic images can be scaled efficiently, they tend to be similar when prompts are fixed. Do retrieved images have more variation with larger data scales than synthetic images? How do the results change if synthetic images in the scaling experiment are sourced using captions from retrieved images?

[1] Image Captions are Natural Prompts for Text-to-Image Models

**Questions:**

Please address the weaknesses.

**Limitations:**

Limitations have been discussed in the paper.

---

> ### Author Rebuttal · Authors · 2024-08-07
>
> Thank you for your valuable feedback! We are excited that you found our proposed baseline for measuring the utility of synthetic data to be “novel and inspiring.”
>
> Your feedback prompted us to perform additional experiments to further validate our paper’s findings, and we believe the new results corroborate our paper’s main finding that state-of-the-art synthetic data training methods lag our proposed retrieval baseline. Thank you for helping us strengthen our paper! We will include all new results in the next version of our paper, and describe the details below to address each of your points.
>
>
> **Q1 (CLIP score distributions post-filtering):** Do the distributions of CLIP similarity scores in the final synthetic and retrieved training datasets significantly differ? Are retrieved and synthetic images ranked and filtered separately, or together?
>
> **A1:** Great question! Significant differences in CLIP similarity could potentially explain gaps in downstream training performance and is worth studying. In our paper, **we ranked and filtered synthetic and retrieved training data separately** (i.e., to keep the experimental design as clean as possible, we do not use any information from the retrieved images to inform the selection of synthetic data, and vice versa). We use per-class score thresholds for filtering, which we found empirically beneficial for both synthetic and retrieved data. We histogram the CLIP similarity score distributions of the resulting filtered training data in Figure R2 of the rebuttal PDF. Overall, despite setting the filter score threshold for synthetic and retrieved data independently, we find that the distribution of post-filtering synthetic image CLIP scores is right-shifted compared to the distribution of post-filtering retrieved image CLIP scores. In other words, **synthetic images have comparable or even higher CLIP scores than retrieved images on average**; CLIP judges synthetic data to be higher quality on average. Thus, CLIP score differences alone do not explain the lagging training performance of synthetic data.
>
> **Q2 (Filtering out images with artifacts):** As shown in Figure 3, synthetic ‘flute’ images have obvious generation artifacts. These images should be filtered for fair comparison.
>
> **A2:** To clarify, the artifact-afflicted images shown in Figure 3 are *post-filtering*. Even though the ‘flute’ images of Figure 3 are obviously wrong to humans, CLIP assigns them relatively high scores of $0.285, 0.265$, and $0.263$. For reference, a CLIP score of $0.249$ reflects the top $30$% of all retrieved ‘flute’ images. Furthermore, we found obviously wrong and artifact-ridden synthetic ‘flute’ images that have even higher CLIP scores $> 0.3$, placing them in the top $5$% of all retrieved ‘flute’ data.
>
> Our paper adopts CLIP filtering since it is the current best synthetic image filter [22] we are aware of; in fact, many recent works forgo data filtering altogether [3,21,51,57,58] despite its positive impact on training performance [22] (our study corroborates this gain; CLIP filtering synthetic data improved ImageNet zero-shot accuracy by 1.03% and LP by 0.91% over no filtering at 4M scale). We aim to ground our argument in the current state of synthetic training data research, so we do not explicitly innovate new data filtering methods beyond CLIP filtering. Nonetheless, these findings spurred by your feedback makes it apparent that CLIP filtering is limited. We will include a more detailed discussion of this point in our final paper to motivate future filtering methods.
>
>
> **Q3 (Alternative prompts for synthetic data generation):** Synthetic images may suffer in diversity when the generation prompts are fixed. Other prompting strategies should be considered, such as using LAION alt-text or outputs from a captioning model.
>
> **A3** Thank you for the point! First, to clarify a potential miscommunication, the prompts used to generate synthetic images in our work are *not fixed* – rather, each image prompt is sampled from a probabilistic large language model (LLM) that is tasked to include additional relevant knowledge about the desired category (e.g., for the “dog” category, a sampled generation prompt might be “a photo of a dog playing with a ball in the park”). We will make this more clear in our paper.
>
> We adopt this LLM-guided prompting strategy from SynCLR [57], which shows that it outperforms many alternative prompting strategies at the hundred-million scale, including prompting with LAION alt-text. We did not find any existing comparison between LLM-generated prompts and prompts from an image captioning model as proposed by the work you referenced [1], so we conducted this study ourselves.
>
> Starting from the unfiltered retrieved training set, we used BLIP-2 to caption each image and construct prompts of the form “a photo of {classname}, {BLIP-2 image caption}” following [1]. We then performed top-30% CLIP filtering on the resulting synthetic images. We compare the performance of training with filtered synthetic images generated with three prompting distinct strategies to our filtered retrieved data in Figure R3 of the rebuttal PDF. Specifically, we compare training with filtered synthetic images generated from (1) our original LLM prompts (orange lines), (2) BLIP-2 captions (red lines), and (3) LAION alt-text from retrieved data (green lines). Overall, among the three generation strategies, our original LLM prompts perform best on ImageNet, and perform comparably to BLIP-2 captions and LAION alt-text on Aircraft. All three synthetic strategies lag the performance of retrieved data. Thus, our conclusion—that existing synthetic image training methods do not surpass the retrieval baseline—holds under these variations of image generation strategy.
>
> We were not aware of [1] at the time of submission — thank you for pointing it out to us! We will include a reference to that work and the above experiments motivated by [1]’s captioning method in our final paper.

---

> > ### Comment · Reviewer_Etje · 2024-08-12
> >
> > Thanks to the authors for the rebuttal. Did I miss the response to my last question, quoted below?
> >
> > > Data scaling: Although synthetic images can be scaled efficiently, they tend to be similar when prompts are fixed. Do retrieved images have more variation with larger data scales than synthetic images? How do the results change if synthetic images in the scaling experiment are sourced using captions from retrieved images?

---

> ### Author Response · Authors · 2024-08-12
> **Thank you for your reply! We apologize for the confusion and are happy to discuss further.**
>
> Thank you for your time and continued interest in our work! We apologize for our confusing formatting — our initial response to your last question is folded into **Q3/A3** of the rebuttal text above. For your convenience, we summarize the relevant parts here:
>
> * First, to clarify a potential miscommunication, our generation prompts are not fixed, but rather sampled from a probabilistic LLM that is tasked to generate image captions for each desired visual concept. We use these LLM captions as text-to-image prompts.
> * We adopt the LLM-guided prompt method from SynCLR [57], a current SOTA work which shows that synthesizing images with LLM prompts can outperform alternative generation strategies (including synthesizing images from LAION alt-text) at hundred-million scale.
> * We experimentally compared the training performance of filtered synthetic images generated using either (a) our original LLM prompts or (b) the LAION alt-text of our retrieved images. We report results of this new scaling experiment in Figure R3 of the rebuttal PDF. Synthetic images generated via LLM prompts outperform images generated with LAION alt-text prompts on ImageNet, and perform comparably on FGVC-Aircraft. Regardless of generation strategy, synthetic images lag the retrieval baseline.
>
> We missed the part of your question about the diversity of retrieved versus synthetic images in our initial response. We sincerely apologize for the confusion! To clarify your question, we further compared the image variation in our final filtered Aircraft and ImageNet adaptation datasets, which consist of either (a) retrieved real images, (b) synthetic images generated with our original LLM prompts, or (c) synthetic images generated from the LAION alt-text of the retrieved images. We quantify image variation via the average pairwise cosine similarity of the CLIP image features for each dataset (i.e. lower average similarity → higher variation). To understand variation at large scale, we perform this analysis on the largest-sized version of each dataset . Results are as follows, with specific scale size in parentheses:
>
> |   | ImageNet-1K | FGVC-Aircraft |
> |---|----------|----------|
> | **Retrieved Real Data** | 0.323 (2.5M images) | 0.506 (139K images) |
> | **Synthetic Data (Original LLM Prompts)** | 0.369 (4M images) | 0.606 (500K images) |
> | **Synthetic Data (LAION alt-text)** | 0.341 (2.5M images) | 0.527 (139K images) |
>
> Overall, your intuition is correct: synthetic images generated with LLM prompts indeed exhibit higher average pairwise cosine similarity compared to retrieved images, suggesting deceased variation. This gap persists even though the scales of the LLM prompt synthetic datasets are significantly larger than the scales of the retrieved datasets (e.g. 4M vs 2.5M for ImageNet, 500K vs 139K for Aircraft). Moreover, generating synthetic images based on the LAION captions of the retrieved images does improve the measured variation of the resulting synthetic images.
>
> However, interestingly, improvements in synthetic image diversity alone do not directly translate into significant improvements in downstream model performance. As shown in Figure R1, models trained on the less-diverse LLM prompt generated images we originally considered perform better or comparably to models trained on the more-diverse LAION alt-text generated images. Nonetheless, we are excited that the experiments you suggested have uncovered another axis along which synthetic and retrieved images differ. We will include these experiments and detailed discussion in the updated version of our paper to help motivate future synthetic data work.

---

> > ### Comment · Reviewer_Etje · 2024-08-13
> >
> > Thanks for the further discussions. I will maintain my score.

---

### Official Review · Reviewer_rgD8 · 2024-07-12

**Soundness:** 2
**Presentation:** 2
**Contribution:** 2
**Rating:** 5
**Confidence:** 3

**Summary:**

The paper tries to answer the question of whether the progress of pretraining classification backbones with images obtained from generative models is due to the advances in generative image modeling or from the fact that these are implicitly sampled from huge image collections. To answer this question, the paper proposes a simple baseline consisting of querying the original databases on which the generative models are trained, finding nearest neighbors for the task at hand, and training on these neighbors instead of the generative samples. The paper shows that this simple baseline outperforms naively sampling from the image models.
Furthermore, the paper analyzes why the generated data underperforms real data, concluding that the downgraded performance is due (at least in part) to a lack of fine-grained detail (e.g. in the case of the FGVC dataset, the generated ) as well as artifacts in the generated images that introduce a domain gap with respect to real images.

**Strengths:**

- The question the paper tries to answer is highly relevant to the vision community (and also for other communities such as language), and pinpoints a clear deficiency of the baselines of papers tackling synthetic data generation with pre-trained models. It is highly positive for the community that the paper raises this concern and checks the actual performance of the proposed baseline when compared against simple ways of training for classification using synthetic images.

- The analysis of what causes the models trained with synthetic images to fail is interesting.

**Weaknesses:**

- The main weakness of the paper is that it points out a deficiency of other papers, but its technical contribution is limited (i.e., nearest-neighbor retrieval). Although the question studied is really relevant to the community and the baseline pinpointed should always be considered in papers, the technical contributions of the paper seem limited.

- Sentences 33-35 are (at minimum) a bit ambiguous due to wording (the "over" in L34 is ambiguous), or they are wrong. By the data processing inequality, the generated samples cannot contain any additional information about the images in LAION-2B that is not contained in the images of LAION-2B themselves. However, the generated images can contain additional information that is not present in the original LAION-2B images, and that is useful for the downstream classification task. In fact, the images generated by the generative neural network also contain information about the neural network architecture and the training algorithm. If this was not the case, the mutual information between the generated images and the training algorithm would be 0, and thus synthetic and real images would be indistinguishable.
The *promise* of using generated images is precisely that training on synthetic images adds crucial extra information to the information present on LAION-2B about regularities of the world that are embedded in the implicit biases of the neural network and its training procedure. The implicit biases of neural networks add extra information about the composability of concepts (e.g. "a cow on the beach" is the same as "a horse on the beach" with the cow replaced by a horse, although there are no cows in the beach during training), stability with respect to small perturbations in both text and image space ("a cow in the beach" and "a cow in an island" are roughly the same) and other useful regularities of the world. I strongly encourage the authors to rephrase this sentence and explain that the gains in [57] are well motivated by this.

- Although interesting, the analysis in Section 5 is limited, and other SOTA image models should be tested. Only using StableDiffusion is well motivated in the experimental section (as it is required to have access to the training dataset), but in the analysis section, stronger public or proprietary models should be analyzed to see if they also suffer from the same deficiencies.

- Similar baselines have already been used in the past, and they are not correctly cited. StableRep [57] implicitly has the baseline proposed in the paper, as it compares against using the complete set of real data used for training the generative model (i.e. all the neighbors), showing that it outperforms it. Similarly, the preliminary work of [1] also compares against using the entire training dataset, showing that in this preliminary case, the generative models do not outperform the real datasets.

- The comparison is just against a simple way to generate synthetic images, although several works have proposed more sophisticated ways of sampling that target the shortcomings studied in Section 5. For example, [2] trains on a particular modality with scarce data, instead of using a large-scale pretrained model (which may not have fine-grained details about all the classes). As seen in Figure 2, training with generative samples underperforms over directly using CLIP, while works like [57] outperform it.

- It would be good to study how close the retrieved samples are to the training splits of the datasets studied. Are the training samples of these datasets included in LAION-2B? If this is the case, and these are properly retrieved, it is expected that the real images are strictly upper bound (as they are perfect samples of the real distribution, and there won't be any distributional shift between training and testing).

Minor comments:
- L176 and Figure 2: The use of the term "Zero-shot" performance to refer to the text-query classification performance is a bit confusing, as the tested models have been finetuned with N-shots. If I understand correctly, this corresponds to applying the zero-shot classification technique of CLIP after finetuning on N samples for each class, but as the models have been finetuned with N-shots.

**Questions:**

See weakness

**Limitations:**

Yes

---

> ### Author Response · Authors · 2024-07-31
> **Clarification request on references [1] and [2]**
>
> Thank you for your valuable review! We highly appreciate your feedback. To ensure that we can thoroughly address the review, could you please clarify what works [1] and [2] refer to? It seems that [1], [2] in the review text do not match references [1], [2] in our paper. Also, to double-check, [57] in our paper is a citation to SynCLR -- does "StableRep [57] implicitly..." in the review text refer to SynCLR, or does it refer to a different citation? Thank you very much for your time!

---

> ### Comment · Reviewer_rgD8 · 2024-07-31
> **References clarification**
>
> [1] Generative models as a data source for multiview representation learning. Ali Jahanian, Xavier Puig, Yonglong Tian, Phillip Isola" (GenRep)
>
> [2] Using diffusion models to generate synthetic labeled data for medical image segmentation. Daniel G Saragih, Atsuhiro Hibi , Pascal N Tyrrell
>
> [3] StableRep: Synthetic Images from Text-to-Image Models Make Strong Visual Representation Learners. Yonglong Tian, Lijie Fan, Phillip Isola, Huiwen Chang, Dilip Krishnan
>
> [57] corresponds to the submission citation (SynCLR). All three GenRep [1], StableRep [3] and SynCLR [57] "implicitly have the baseline proposed in the paper", by using the entire dataset (i.e. all the neighbors). In the three papers, the networks are pretrained with generative samples, and the full datasets are used to train the generative models. [1] does not outperform the baseline, while [3] and [57] do.

---

> ### Author Rebuttal · Authors · 2024-08-07
>
> Thank you for your valuable review! We are glad you found our research question “highly relevant” for both vision and NLP.
>
> Before answering your points directly, we’d like to clarify a possible miscommunication. A key point in the review is that our proposed retrieval baseline already implicitly exists in other work. For example, SynCLR [57] compares synthetic images against the generative model’s full training set (i.e. retrieve everything) and finds that synthetic data performs better.
>
> While such results are exciting, this existing baseline—comparing synthetic data to the generator’s full train set—is distinct from our proposed baseline, and is insufficient for measuring the true added value from the generative model itself. When we sample synthetic training data, we often sample data that is targeted to tasks we want our learner to perform well. Works like [57] use this idea to generate synthetic data that beats the full real dataset for specific tasks. However, by comparing to the full real data, a task-agnostic distribution of images, [57] and other works [21] implicitly conflate the effects of training on synthetic versus real data with the effects of targeted data sampling. We aim to disentangle these factors and answer: are observed gains from synthetic data truly due to added information from the generator? Or is it since the synthetic data is implicitly sampled in a targeted manner from large upstream real datasets? To study this, we construct a retrieval baseline that explicitly controls for targeting (L39-45), thus overcoming a key limitation of prior work.
>
> Overall, we are excited by the same possibility you highlighted—that synthetic data can transfer useful information from the generator's implicit biases. However, to ensure our field progresses toward this goal, we must carefully measure the current state of synthetic data methods. We believe our work is a crucial step in this direction.
>
> **Q1:** “The main weakness of the paper is that… Although the question studied is really relevant to the community… the technical contributions of the paper seem limited.”
>
> **A1:** Our goal is not to innovate new methods; rather, we seek to introduce a conceptual baseline that helps our field better understand the true utility of synthetic training images. Our proposed baseline can be implemented with existing image retrieval techniques; we aim to keep our baseline simple to facilitate comparison against it.
>
> **Q2:** Unclear wording of data processing inequality sentence.
>
> **A2:** Thanks! We will revise to discuss that SynCLR [57] is indeed motivated. We agree that synthetic images can contain useful information from the generator’s training process. However, it is unclear if [57]'s gains are truly due to this added information—[57]'s gains may also be from targeting the synthetic data to the evaluation tasks. Our retrieval baseline controls for the effect of targeting, allowing us to better measure whether the observed gains from synthetic data are due to the generator’s added information.
>
> **Q3:** Why only analyze Stable Diffusion (SD), why not analyze closed-source models to see if they suffer similar limitations?
>
> **A3:** We analyze SD so we can contextualize its analysis with its performance relative to our retrieval baseline, which is only possible with open data. In contrast, if we find some closed-source generative model G yields higher quality images than SD, it would be unclear whether these gains in image quality reflect improvements in the information added by G to the synthetic images, or whether G was simply trained on a higher quality dataset than SD. We also aim to keep consistency with recent work [21,51,57] that all use SD as the generator. We will revise to clarify these points.
>
> **Q4:** The proposed baseline exists in prior works, which compare synthetic data against the generator’s full training set.
>
> **A4:** Please see our discussion above for a detailed response. Briefly, SynCLR [57] does not control for the effects of data targeting, and GenRep [1] samples data from unconditional GANs, whereas modern synthetic data is sampled in a targeted manner. Our retrieval baseline which controls for targeting is now necessary to understand synthetic data gains in the modern regime. StableRep [3] only compares against a small random subset of the full train data. We will clarify in our final paper. Thank you!
>
> **Q5:** The comparison is against a simple way to generate synthetic training images, but works like SynCLR [57] propose more advanced ways that can outperform CLIP.
>
> **A5:** To clarify, our paper exactly adopts [57]’s synthetic data method. [57] only significantly beats CLIP on the Aircraft task, which our results corroborate. Sampling data from generators trained on data-scarce domains as you cited [2] is an exciting idea that we will discuss in related work; we focus our study on sampling data by prompting web-pretrained models like Stable Diffusion to be consistent with other work [21,51,57].
>
> **Q6:** The high performance of retrieved data may be due to training set contamination.
>
> **A6:** Good point! The generator is trained on the same data we retrieve from, so in theory synthetic data can also have benchmark train set contamination. We agree this contamination is more direct when we retrieve data; we thus further decontaminated all retrieved data for the train sets following [18]. We report the amount of removed data in Table R1 and plot the results of training on decontaminated retrieved data in Figure R1 of the rebuttal PDF. Overall, while some retrieved images were indeed similar to train set data (an average 1.9% of retrieved data was removed), discarding them minimally impacted performance.
>
> **Q7:** Confusing zero-shot terminology.
>
> **A7:** Thanks! We will clarify in our paper: our zero-shot models are not trained on downstream benchmark data. We finetune CLIP on retrieved or synthetic data only and evaluate the resulting model as-is on the test set.

---

> ### Comment · Reviewer_rgD8 · 2024-08-09
> **Thanks for the rebuttal**
>
> Thanks for the rebuttal. Could authors give a bit more details about what they mean by targeted refinement and why  [57] and [21] "implicitly conflate the effects of training on synthetic versus real data with the effects of targeted data sampling". I still struggle to see the difference between [57] and "targeted refinement", is it simply that [57] targets the full training dataset instead of a subset like the case studied? Why is this substantially different?

---

> > ### Author Response · Authors · 2024-08-12
> > **Thank you for your response! We are happy to clarify and discuss further.**
> >
> > Thank you for your reply! We truly appreciate your time and continued interest. To illustrate what we mean by prior work “conflating targeted data sampling with synthetic data,” we detail why SynCLR's experiments [57] cannot answer our research question.
> >
> > Our work asks: does synthetic training data contain useful added information beyond the training dataset of its generator? We believe this is timely as recent works (e.g. [57]) have shown that training on a synthetic dataset $D_S$ can outperform training on the full real dataset $D_R$ used to pretrain its generator. However, this prior finding—that $D_S$ outperforms full $D_R$—does not answer our question, as the experimental setup confounds two distinct independent variables. To see the confounded variables, let's recap SynCLR's method for generating a synthetic training dataset $D_S$:
> >
> > 1. Manually define a set of visual concepts that we want our model to perform well on. SynCLR's concept list is largely based on the classes in the main downstream evaluation tasks (Tables 6,11 in [57]). For example, since we want to perform well on FGVC-Aircraft and ImageNet, the concept set contains ‘Airbus A320’, ‘flute’, etc.
> > 2. Use an LLM to generate image captions for each concept.
> > 3. Generate an image for each caption via Stable Diffusion, a text-to-image model trained on LAION-2B.
> >
> > Thus, $D_S$ is collected with so-called “targeted data sampling:” the synthetic dataset $D_S$ SynCLR trains on isn’t sampled unconditionally and is not intended to cover the full LAION-2B distribution. Rather, it is carefully tailored to specific tasks based on the manually-fixed concepts in step 1. As such, the synthetic dataset $D_S$ differs from LAION-2B, the real training set of Stable Diffusion (i.e. $D_R$), along two axes simultaneously. First and most apparent, (a) LAION-2B is real while (b) $D_S$ is sampled from a generator. More subtly but just as critical, (i) $D_S$ is targeted to the downstream eval tasks, while (ii) LAION-2B is a broad, task-agnostic distribution of images. To illustrate, $D_S$ is constructed such that 5% of the dataset is images of FGVC-Aircraft classes (30M total images). In contrast, we estimated the proportion of FGVC-Aircraft relevant images in LAION-2B to be just 0.056% (only ~1.1M total images). Thus, while SynCLR shows that training on $D_S$ outperforms training on full LAION-2B for FGVC-Aircraft and performs comparably on other benchmarks, any gains $D_S$ exhibits over full LAION-2B may not be because $D_S$ is generated while LAION-2B is real. Rather, it could simply be because full LAION-2B is task-agnostic, while $D_S$ is task-targeted. Importantly, whether data is targeted is not a unique property of synthetic data—real data can be targeted too (e.g. via retrieval as discussed below). With existing baselines, we cannot attribute the gains from training on $D_S$ to Stable Diffusion adding unique value on top of its LAION-2B training set; we cannot resolve our research question.
> >
> > The conflation of these two independent variables (data can be synthetic or real; data can be targeted or untargeted) is not unique to SynCLR. Modern synthetic data methods often rely on prompting text-to-image models; synthetic data is thus often targeted via the generation prompts. For example, SynthCLIP [21] derives prompts from the MetaCLIP concept list, thus targeting synthetic data to MetaCLIP concepts. SynthCLIP also compares this targeted synthetic data to general real data (e.g. Conceptual Captions).
> >
> > The key innovation of our retrieval baseline is to enable disentangling these presently conflated variables, thus creating a more principled experimental setup. Through retrieval, we collect targeted real data from LAION-2B and compare targeted synthetic data $D_S$ head-to-head against this targeted real data. Our new setup now only varies one independent variable (i.e. synthetic versus real). Any gains $D_S$ exhibits over our retrieval baseline are thus properly attributable to $D_S$ having added information over the generator’s training set. However, we show that even SOTA synthetic data methods (we adopt SynCLR) lag the retrieval baseline. When we correctly control for synthetic data being targeted, previously observed gains vanish. In summary:
> >
> > * Works like SynCLR compare targeted synthetic data to the generator's full untargeted real training set. This controls for the effect of seeing information from generator training data, but does not control for the equally critical effect of targeted data collection.
> > * We point out this gap and propose the retrieval baseline to further control for data targeting and resolve it.
> > * We show that gains from SOTA synthetic methods vanish against this principled baseline.
> >
> > Our work contributes a way to measure if synthetic data has useful information beyond the generator's training set, thus yielding a clean target for future methods. We're eager to see the gains our field will find.
> >
> > Thank you! We are happy to discuss further!

---

> ### Comment · Reviewer_rgD8 · 2024-08-12
> **Thanks for the clarification**
>
> Thanks for the detailed clarification, you are right! If I understand correctly, and as specified in Section 4.1, SynCLR does create a LAION-like set of captions, and they train on all the captions once (using CLIP), and not retrain per each downstream task. Although this is less severe than training specifically for each downstream application (i.e. only training on Airplanes when the downstream is also Airplanes), it is true true that during the process of creating captions they use all the captions from all downstream tasks (with many variations and augmentations), so indeed, their process conflates the effects of the synthetic sampling and the targeting of (all) the downstream tasks they study.
>
> After the discussions, authors have addressed my concerns, and I'm raising my score to acceptance. Still, I expect the final version of the manuscript to include clarifications on the mentioned points.

---

> > ### Author Response · Authors · 2024-08-12
> > **Thank you for your time!**
> >
> > Thank you very much for your time and effort in this process! We enjoyed discussing our work with you, and are glad that we were able to address your concerns. We appreciate your score increase. We will clarify based on the discussed points in the next version of our paper. Thank you for helping us strengthen our work!

---

### Official Review · Reviewer_ZHX8 · 2024-07-16

**Soundness:** 3
**Presentation:** 4
**Contribution:** 2
**Rating:** 5
**Confidence:** 3

**Summary:**

This paper evaluates the performance of training machine learning models on synthetic images generated by the Stable Diffusion generative model compared to using real images directly retrieved from the LAION-2B dataset, which was used to train the generative model. The authors argue that while synthetic images can benefit some downstream tasks, they are generally outperformed by directly using targeted real images. The study involves extensive empirical testing across several visual recognition tasks, showing that real data often provides better accuracy and fewer artifacts than synthetic alternatives.

**Strengths:**

(1) This paper innovatively proposes the impact of synthetic data and the corresponding training pairs on downstream tasks, providing a comprehensive analysis by comparing the performance of models fine-tuned on synthetic and real data across multiple tasks and scales. It recommends using true data from the same prompt domain as a baseline for downstream tasks.

(2) The experimental design includes a robust setup, featuring a variety of benchmarks that support the derived conclusions. The authors test their hypotheses across multiple data sizes and report the results with detailed statistical data.

(3) Discussions on the limitations of synthetic data and the potential for real data to yield better training outcomes have a direct impact on the development approaches of future artificial intelligence models, especially in fields where data quality is critical.

**Weaknesses:**

(1) The author aims to emphasize the importance of the original training data as an evaluation baseline, but has not yet compared the downstream fine-tuning results of the real-and-synthetic data of non-training data.  And not clear about the advantage of using upstream real data as baseline but not the othe dataset.

(2) A lack of quantitative indicators to evaluate the quality of synthetic images, the authors only use CLIP for screening. For the generated image data, no more robust quantitative indicators are used to screen and eliminate images with low quality or obvious visual semantic noise. Because in most downstream methods based on synthetic data, generated images are usually screened by a lot of manual and computational methods to improve the fine-tuning quality.

(3) Due to privacy restrictions, model open source and other issues, it is difficult to retrieve training data to compare with synthetic data, which may limit the future application of this work.

**Questions:**

There is a main confuse that the key is "retrieved data - synthetic data" or "high quality data - low quality data" in this field, and the contribution to use retrieved real data as baseline should be more clear and specific. The inspiration this work provides for future work needs to be further clarified by the author, which is also the key to my rating.

**Limitations:**

The conclusions and exploration presented in this work is solid, but the fact that the original training data is of higher quality than the data generated by the generator is not a valuable conclusion.

---

> ### Author Response · Authors · 2024-08-04
> **Clarification request on weakness point (1)**
>
> Thank you very much for your time and effort in providing us feedback! We highly appreciate your review. Could you please clarify your point (1) under the weakness section? We would love to address your points as throughly as possible. Specifically, we were hoping to clarify the following:
>
> *"... downstream fine-tuning results of the real-and-synthetic data of non-training data. And not clear about the advantage of using upstream real data as baseline but not the othe dataset."*
>
> By "non-training data," do you mean that data outside the generative model's training dataset should also be used as the baseline? Does the main question under the Question section ("There is a main confuse...") refer to the same point? Thank you so much for your time!

---

> > ### Comment · Reviewer_ZHX8 · 2024-08-05
> > **Clarification on Weakness (1)**
> >
> > I am glad to rephrase my concern in Weakness (1). I am curious about how text-image pairs from data outside the training set serve as the real-data baseline, compared with the results of the text prompt and generated images. If the result still shows that the quality of the generated data is inferior to real-data, it seems that other real-data from non-training dataset could also serve as a "baseline" for the claims in this article. In that case, what is the necessity of obtaining training data as declared in the article?
> >
> > The question in the Question section revolves around whether the core answer to the performance on downstream CV tasks discussed in the article is "training and generated data" or "high-quality and low-quality data."?
> >
> > Thanks, hope my clarification will help the authors' further response.

---

> ### Author Rebuttal · Authors · 2024-08-07
>
> Thank you for your valuable feedback! We are glad you found our experimental setup “robust” and our discussion to “have a direct impact on the development” of future models. We address each of your points below.
>
> **Q1 (Why retrieval baseline?):** Why emphasize the importance of retrieving data from the generative model’s training data as an evaluation baseline for synthetic data? Why adopt the retrieval baseline over simply comparing synthetic data to high-quality data outside the generative model’s training set?
>
> **A1:** Great question! Our research goal is not just to study whether synthetic training data can improve model performance (which as you noted, would not require the retrieval baseline). Rather, we seek to understand where performance gains from synthetic data come from, and to contribute a principled way of measuring whether model-generated synthetic data can surpass the real data it derives from (i.e., the data used to train the generator). To study this question empirically, we propose the retrieval baseline. This baseline enables us to disentangle whether gains from synthetic data are due to (a) the fact that synthetic images are implicitly subsampled from the generator’s huge real training set, or (b) due to the generator truly adding useful information beyond its training data. If we compare synthetic data to high-quality data outside the generator’s training set, then these two factors cannot be disentangled. For example, even if a synthetic data method outperforms high-quality outside data, the gains from synthetic data may simply be due to the generative model seeing higher-quality and larger-scale data during pretraining. In contrast, our baseline of retrieving real data from the generator’s training set controls for the effect of information from the upstream real data. Retrieval also controls for the effects of targeted data sampling (i.e., synthetic data is often targeted to specific tasks), which has been conflated with synthetic data in prior work [57].
>
> We believe the research questions we pose—how to measure if/when synthetic data surpasses the real data it was trained on—are timely to tackle. There is surging interest in building SOTA vision models with synthetic data [3,21,51,57,58], and many works now show strong gains. We find that such gains—while exciting—still fall short of our retrieval baseline, suggesting that today’s synthetic data methods do not provide significant information beyond the generator’s training data. Hence, we hope our baseline will set a strong and simple target for the field going forward. We further discuss how our paper can impact future work in response to your **Q4** below. We will clarify all points in our final paper. Thank you for helping us strengthen our work!
>
> **Q2 (Insufficient filtering):** The CLIP score used in the paper to filter synthetic images is not sufficiently sophisticated and does not reflect current best filtering practices.
>
> **A2:** Thanks for the question! This point may be mistaken. While we agree that improving data filtering is an exciting future direction for improving the utility of synthetic training data, filtering based on CLIP score remains state-of-the-art [22]. In fact, many recent representative works do not apply any quality filters whatsoever to generated synthetic training images [3,21,51,57,58]. Our work uses CLIP filtering to optimize the performance of synthetic data, which [22] finds improves training performance over no filtering. Our study corroborates [22]’s finding; for example, applying CLIP filtering to synthetic training data improved ImageNet zero-shot and LP accuracy by 1.03 and 0.91 points (respectively) over no filtering at 4M data scale.
>
> **Q3 (Retrieval is impractical):** Retrieval from a generator’s training data is often not practical due to privacy concerns or closed-source data, which may limit the work’s applicability.
>
> **A3:** Good point! We make a similar note and discuss its implications for future research in the introduction and discussion of our paper (L70-73, L312-315). While privacy concerns and closed data are practical reasons to use synthetic data over retrieved data, these application constraints are orthogonal to our research question. Specifically, our goal in proposing the retrieval baseline is not to prescribe a general method for maximizing downstream task accuracy (which, as you noted, would require consideration of downstream constraints), but rather to advance our understanding of what value synthetic data provides beyond the training set of its generator.
>
> **Q4:** How can this research and the proposed retrieval baseline inspire future work?
>
> **A4:** We believe our work can inspire future research in two primary ways. First, comparison against our retrieval baseline provides a principled target for future synthetic data research to aim for. If any synthetic data method outperforms training on data retrieved from the generative model’s training data, then that is strong evidence that the generative model has added (e.g., through its inductive bias, through its compositional abilities) additional useful information on top of the data that it was originally trained on. In contrast, as discussed in **A1**, if a synthetic data method outperforms some other source of real data outside of the generative model’s training set, then we cannot draw a similar conclusion.
>
> Second, by demonstrating that retrieving real data from the generator’s training set currently surpasses synthetic data, we hope to sharpen the field’s intuition of when existing synthetic data methods can be useful in the present day. Many recent works position synthetic data as a potential drop-in replacement for real data, even when the retrieval baseline is possible. Our findings do not corroborate such positioning; instead, existing synthetic data methods are most promising when downstream constraints (e.g., privacy concerns) prevent the retrieval baseline from being realized.

---

> > ### Comment · Reviewer_ZHX8 · 2024-08-12
> > **Thanks for the Rebuttal**
> >
> > Thank you for your rebuttal. Your reply has addressed my concerns regarding Q2 and Q3.
> >
> > Regarding your statement in A4, 'Many recent works position synthetic data as a potential drop-in replacement for real data, even when the retrieval baseline is possible,'. In my opinion,  the main advantage of synthetic data lies in its ability to scale effectively. By leveraging larger datasets, it can enhance performance in other tasks. From this perspective, whether considering data scaling capabilities or privacy risks, synthetic data holds more practical value than retrieving real data.
> >
> > Of course, the exploration of data quality is currently a major issue. The real focus of the field is how to make synthetic data superior to the real data retrieved in the training set not just using what kind of data as baseline. I appreciate your writing and experiments, could this work give us more deeper insights on how to make synthetic data better?

---

> > > ### Author Response · Authors · 2024-08-13
> > > **Thank you for your response! We are excited to discuss further.**
> > >
> > > Thank you for your time and continued interest in our work! We are glad that we were able to address parts of your concern and are very grateful to see that you’ve increased your score in favor of acceptance.
> > >
> > > Overall, we are excited by the same possibility that you highlighted: that synthetic data can be effectively scaled, and that training on ever-larger synthetic datasets can improve our models. However, this goal remains highly non-trivial. Our work found that even when we scale up the amount of synthetic data generated from a current SOTA method [57] beyond the amount of retrieved data considered, synthetic data still lags retrieved data in terms of downstream training performance (**Figure 1, L199-220**). For example, training on a mere 30K retrieved images outperformed 500K synthetic images for FGVC-Aircraft. We further find that with current methods, scaling up synthetic data can often hurt performance, not help. Thus, we fully agree that scaling up synthetic data has exciting potential, but retrieved data is currently more performant than synthetic data when available. To realize the untapped potential of synthetic data, we need to find better methods. And to build better synthetic data methods, our field needs a principled target to aim for.
> > >
> > > We believe that our retrieval baseline may be one such target. If any synthetic data method outperforms training on data retrieved from the generative model’s training set, then that strongly suggests that the gains come from improvements in our synthetic data methodology itself, as opposed to improvements in the generator’s real training data. In the second case, using the generator’s real training dataset directly would continue to be more practical when available; synthetic data would remain primarily useful under privacy constraints.
> > >
> > > While privacy is already an exciting application domain, we are optimistic about future improvements in our synthetic data methods to make them useful more broadly (i.e. even when the generator’s real train set is available). To this end, we also performed analysis experiments (**Section 5.1, Figures 3,4**) to better understand why synthetic data currently lags retrieved data. We show that both visual artifacts from the generator and high-level semantic content differences contribute to the underperformance of synthetic data, thus highlighting these two axes as future directions for improving synthetic data. Moreover, by conceptualizing retrieval as a target to beat, another natural future step could be to design methods that generate image compositions which are explicitly absent from the generator’s upstream training set; synthesizing these “missing" images may offer unique value beyond the existing upstream real images. Such an approach leverages the compositional generalization abilities of generative models, which recent research promisingly suggests may be a unique boon of generative models compared to other models trained on the upstream data [1, 2] (references below). We will thoroughly discuss these other ways in which our work may motivate future synthetic data improvements in our updated paper. Thank you for your thoughtful questions!
> > >
> > > Thank you again for your time and valuable feedback, which has improved our work. We share your excitement about the potential of synthetic data, and are optimistic that our work will inspire subsequent works to find gains beyond our retrieval baseline. We would be eager to discuss further!
> > >
> > > * [1] Your diffusion model is secretly a zero-shot classifier. Alexander C Li, Mihir Prabhudesai, Shivam Duggal, Ellis Brown, and Deepak Pathak.
> > > * [2] Text-to-image diffusion models are zero shot classifiers. Kevin Clark and Priyank Jaini.

---

### Author Rebuttal · Authors · 2024-08-07

We sincerely thank all reviewers for their detailed feedback. We will incorporate all suggestions in the next version of our paper. Thank you all very much for your invaluable help in improving our work!

Overall, we are thrilled to see that all reviewers found the research question we posed interesting and relevant to the community. The two main concerns were raised by **reviewers ZHX8 and rgD8**, who wondered if the retrieval baseline proposed in our paper is a necessary contribution as opposed to simpler and existing alternatives.

For example, **reviewer ZHX8’s** primary concern questioned why we might want to compare synthetic data to data specifically retrieved from the generative model’s training set, as opposed to a simpler baseline of general high-quality real data that need not come from this specific source. We detail why this simpler baseline would not allow us to answer our research questions in response to the reviewer below. We will also clarify in our revised paper. To summarize, our goal is not just to understand whether training on synthetic data can improve model performance (in which case the simple baseline would suffice), but rather to better understand where any gains we observe derive from. Do gains from synthetic data come from the fact that we are implicitly subsampling relevant data from the generator’s huge real training set (which we can alternatively do via retrieval)? Or do gains come from the generative model truly adding some new information (e.g., through its inductive biases) that surpasses its training data? Comparison against the simple baseline cannot disentangle these two factors; any observed synthetic data gains versus the simple baseline may simply arise from improvements in the generative model’s training data quality. As interest in building SOTA vision models with synthetic data [3, 21, 51, 57, 58] is surging, we believe these are timely unanswered questions to tackle. Hence, we propose the retrieval baseline to enable studying them empirically.

**Reviewer rgD8** expressed concern that our retrieval baseline already implicitly exists in previous work like SynCLR [57], which has compared synthetic data to the *full* training set of the generative model. We detail why this baseline is also distinct from our baseline in response below. Mainly, the full dataset baseline does not control for the critical effect of data targeting, which has been implicitly conflated with synthetic data in past work [57] (L39-45). We show that by comparing synthetic data against our baseline, previous gains shown from synthetic data—while still exciting—largely go away. Thus, we believe our baseline is critical for understanding the true added value of synthetic training data.

**Reviewer rgD8 and reviewer Etje** both asked for additional empirical experiments to further validate the main findings of our paper. We have performed the requested experiments, and include all resulting figures and tables in the rebuttal PDF. We summarize the new experiments here:

* Train set decontamination (Figure R1, Table R1). **Reviewer rgD8** pointed out that retrieving data from LAION-2B may result in images from the benchmark training set (e.g., from ImageNet) being included in the retrieved data. Our retrieved sets are already decontaminated for the downstream test set; we thus performed additional train set decontamination and plotted the results of training on the train+test decontaminated data. Overall, train set decontamination has minimal impact on model performance and does not change our findings.

* CLIP score distributions (Figure R2). **Reviewer Etje** wondered whether there were significant discrepancies in the CLIP score of post-filtering synthetic and retrieved images. We find that if anything, CLIP judges synthetic data to be higher quality than retrieved data on average, despite its lagging training performance. We are optimistic that this new experiment will motivate future synthetic data filtering methods that do not rely on CLIP.

* Additional synthetic image prompt strategies (Figure R3). **Reviewer Etje** wondered whether generating synthetic images with other prompting strategies would yield higher training performance. We compare our original LLM-guided prompting strategy to prompts from the LAION alt-text or BLIP-2 captions of our retrieved data, and find that the original LLM-guided strategy performs better on ImageNet and similarly on FGVC-Aircraft compared to the two alternatives.

We believe the new experiments further corroborate our main findings, and we will include all of them in the next version of our paper. Once again, thanks to all the reviewers and chairs whose effort makes this process possible. We deeply appreciate your effort in helping improve our work! We are eager to further discuss with all reviewers during the discussion period.

---

### Decision · Program_Chairs · 2024-09-25

**Decision:**

Accept (poster)

**Comment:**

This paper evaluates the effectiveness of training machine learning models using synthetic images generated by the Stable Diffusion generative model compared to real images directly retrieved from the LAION-2B dataset, which was used to train the generative model. The authors demonstrate that, while synthetic images can be beneficial for certain downstream tasks, real images generally outperform them, offering better accuracy and fewer artifacts. The study is backed by extensive empirical testing across various visual recognition tasks.

The paper addresses a timely and important question about the role of synthetic data in model training and proposes a strong baseline for future work in this area. Although the technical contributions are somewhat limited, the insights provided are valuable and likely to shape future research directions. Reviewers generally agree on accepting the paper, acknowledging its solid methodology and potential impact despite some noted weaknesses. Therefore, the Area Chair recommends acceptance.

The following suggestions are recommended for inclusion in the final version: 1) exploring scenarios where synthetic data may have advantages, such as in handling rare or complex concepts; 2) investigating the combination of synthetic and real data, and 3) improving the explanation of the contributions of retrieved real data as a baseline.